# Axonal distribution of mitochondria maintains neuronal autophagy during aging via eIF2β

**Kanako Shinno[1†], Yuri Miura[2], Koichi M Iijima[3,4], Emiko Suzuki[1,5], Kanae Ando[1,6]\***

[1]Department of Biological Sciences, Graduate School of Science, Tokyo Metropolitan University, Hachioji, Japan; [2]Research Team for Mechanism of Aging, Tokyo Metropolitan Institute for Geriatrics and Gerontology, Itabashi, Japan; [3]Department of Neurogenetics, National Center for Geriatrics and Gerontology, Obu, Japan; [4]Department of Experimental Gerontology, Graduate School of Pharmaceutical Sciences, Nagoya City University, Nagoya, Japan; [5]Gene Network Laboratory, National Institute of Genetics and Department of Genetics, SOKENDAI, Mishima, Japan; [6]Department of Biological Sciences, School of Science, Tokyo Metropolitan University, Hachioji, Japan

**\*For correspondence:**
k_ando@tmu.ac.jp

**Present address:** [†]Department of Neuroscience and Pathobiology, Research Institute of Environmental Medicine, Nagoya University, Nagoya, Japan

**Competing interest:** The authors declare that no competing interests exist.

## eLife Assessment

In flies defective for axonal transport of mitochondria, the authors report the upregulation of one subunit, the beta subunit, of the heterotrimeric eIF2 complex via mass spectroscopy proteomics. Neuronal overexpression of eIF2β phenocopied aspects of neuronal dysfunction observed when axonal transport of mitochondria was compromised. Conversely, lowering eIF2β expression suppressed aspects of neuronal dysfunction. While these are intriguing and **useful** observations, technical weaknesses limit the interpretation. On balance, the evidence supporting the current claims is suggestive but **incomplete**, especially concerning the characterization of the eIF2 hetero-trimer and the data regarding translational regulation.

**Abstract** Neuronal aging and neurodegenerative diseases are accompanied by proteostasis collapse, while the cellular factors that trigger it have not been identified. Impaired mitochondrial transport in the axon is another feature of aging and neurodegenerative diseases. Using *Drosophila*, we found that genetic depletion of axonal mitochondria causes dysregulation of protein degradation. Axons with mitochondrial depletion showed abnormal protein accumulation and autophagic defects. Lowering neuronal ATP levels by blocking glycolysis did not reduce autophagy, suggesting that autophagic defects are associated with mitochondrial distribution. We found that eIF2β was increased by the depletion of axonal mitochondria via proteome analysis. Phosphorylation of eIF2α, another subunit of eIF2, was lowered, and global translation was suppressed. Neuronal overexpression of *eIF2β* phenocopied the autophagic defects and neuronal dysfunctions, and lowering *eIF2β* expression rescued those perturbations caused by depletion of axonal mitochondria. These results indicate the mitochondria-eIF2β axis maintains proteostasis in the axon, of which disruption may underlie the onset and progression of age-related neurodegenerative diseases.

## Introduction

Neurons have a morphologically complex architecture composed of microcompartments and require tight regulation of the abundance of proteins and organelles spatially and temporally (*Yerbury et al., 2016*). Such control of protein amounts, or proteostasis, is essential for neuronal functions (*Hetz, 2021*) and is achieved through the orchestration of protein expression, folding, trafficking, and degradation controlled by intrinsic and environmental signals (*Balch et al., 2008*). Translation is initiated by the eukaryotic initiation factor 2 (eIF2) complex (*Kimball, 1999*). eIF2, a heterotrimer of α, β, and γ subunits, transports Met-tRNA to the ribosome in a GTP-dependent manner (*Jackson et al., 2010*). Under stressed conditions, phosphorylation of eIF2α attenuates global translation and initiates translation of mRNAs related to the integrated stress response (ISR) (*Pakos-Zebrucka et al., 2016*). As for protein degradation, autophagy and proteasome are major systems that maintain proteostasis (*Kroemer et al., 2010*). The proteasome degrades unnecessary proteins, followed by regulated ubiquitination processes (*Nandi et al., 2006*), and autophagy removes damaged or harmful components, including large protein aggregates and organelles, through catabolism (selective autophagy) (*Glick et al., 2010*). In addition to autophagy induced by acute stressors, a basal level of selective autophagy mediates the global turnover of damaged proteins (*Vargas et al., 2023*).

Such constitutive autophagy decreases during aging, which may underlie declines in the structural and functional integrity of neurons (*Aman et al., 2021*). Decreased protein degradation and accumulation of abnormal proteins also contribute to increased risks of neurodegenerative diseases. Age-related neurodegenerative diseases such as Alzheimer's disease and Parkinson's disease are often associated with the accumulation of misfolded proteins such as amyloid-β, tau, and α-synuclein (*Ross and Poirier, 2004*). Enhancement of autophagy mitigates age-related dysfunctions and neurodegeneration caused by proteotoxic stress (*Rubinsztein et al., 2011*). However, it is not fully understood how aging disrupts the regulation of this constitutive autophagy.

Neurons are also highly energy-demanding cells. At nerve terminals, action potentials trigger the release of neurotransmitters via exocytosis of synaptic vesicles, which requires a constant supply of ATP and calcium buffering (*Vos et al., 2010*). Such neuronal activity relies on mitochondrial functions (*Cheng et al., 2010*), and mitochondria are actively transported from their major sites of biogenesis in soma to axons (*Hollenbeck and Saxton, 2005*). However, the axonal transport of mitochondria declines during aging (*Takihara et al., 2015*; *Milde et al., 2015*; *Vagnoni et al., 2016*). Reduced axonal transport of mitochondria is thought to contribute to age-related declines in neuronal functions (*Takihara et al., 2015*, *Vagnoni et al., 2016*; *Morsci et al., 2016*; *Adalbert and Coleman, 2013*). The number of functional mitochondria in synapses is reduced in the brains of patients suffering from age-related neurodegenerative diseases such as Alzheimer's disease (*Duncan and Goldstein, 2006*), and mutations in genes involved in mitochondrial dynamics are linked to neurodegenerative diseases (*Chen and Chan, 2009*). The mislocalization of mitochondria is sufficient to cause age-dependent neurodegeneration in *Drosophila* and mice (*Iijima-Ando et al., 2012*; *López-Doménech et al., 2016*), indicating that the proper distribution of mitochondria is essential to maintain neuronal functions. Thus, depletion of functional mitochondria from axons and proteostasis collapse are common features of aging and neurodegenerative diseases.

Mitochondrial transport is regulated by a series of molecular adaptors that mediate the attachment of mitochondria to molecular motors (*Hollenbeck and Saxton, 2005*). In *Drosophila*, mitochondrial transport is mediated by milton and Miro, which attaches mitochondria to microtubules via kinesin heavy chain (*Guo et al., 2005*; *Glater et al., 2006*). In the absence of milton or Miro, synaptic terminals and axons lack mitochondria, although mitochondria are numerous in the neuronal cell body (*Stowers et al., 2002*). We previously reported that RNAi-mediated knockdown of *milton* or *Miro* in neurons causes a reduction in axonal mitochondria, age-dependent locomotor defects (*Iijima-Ando et al., 2009*), and age-dependent neurodegeneration in neuropile area starting around 30 days after eclosion (day-old; *Iijima-Ando et al., 2012*), and enhances axon degeneration caused by human tau proteins (*Iijima-Ando et al., 2012*), suggesting that these flies can be used as a model to analyze the effect of depletion of axonal mitochondria during aging. In this study, we investigated a causal relationship between mitochondrial distribution and neuronal proteostasis by using neuronal knockdown of *milton*. We found that depletion of axonal mitochondria reduced autophagy and increased the accumulation of aggregated proteins in the axon prior to gross neurodegeneration. Proteome analysis and follow-up biochemical analyses revealed that neuronal knockdown of *milton* increased eIF2β

levels and lowered phosphorylation of eIF2α in the axon. In addition, *milton* knockdown suppressed global translation. Overexpression of *eIF2β* was sufficient to decrease autophagy and induce neuronal dysfunction, and genetic suppression of *eIF2β* restored autophagy and improved neuronal function in the *milton* knockdown background. These findings suggest that loss of axonal mitochondria and elevated levels of eIF2β mediate proteostasis collapse and neuronal dysfunction during aging.

## Results

### Depletion of axonal mitochondria by knockdown of *milton* or *Miro* causes protein accumulation in the axon

In *Drosophila*, mitochondrial transport is mediated by milton and Miro, which attach mitochondria to microtubules via kinesin heavy chain (*Guo et al., 2005*; *Glater et al., 2006*; *Figure 1A*). It has been reported that expression of *milton* RNAi in neurons via pan-neuronal elav-GAL4 driver reduced milton protein levels in *Drosophila* head lysate to 40% and mito-GFP signals in axons to 50% (*Iijima-Ando et al., 2012*; *Iijima-Ando et al., 2009*).

To test how loss of axonal mitochondria affects proteostasis in neurons, we first examined the accumulation of ubiquitinated proteins. At 14 days old, more ubiquitinated proteins were deposited in the brains of *milton* knockdown flies than in those of age-matched control flies (*Figure 1B*, p<0.005 between control RNAi and *milton* RNAi). There was no significant increase in ubiquitinated proteins in *milton* knockdown flies at 1 day old, suggesting that the accumulation of ubiquitinated proteins caused by *milton* knockdown is age-dependent (*Figure 1—figure supplement 1*). We also analyzed the effect of the neuronal knockdown of *Miro*, a partner of milton, on the accumulation of ubiquitin-positive proteins. Since severe knockdown of *Miro* in neurons causes lethality, we used UAS-*Miro* RNAi strain with low knockdown efficiency, whose expression driven by elav-GAL4 caused 30% reduction of *Miro* mRNA in head extract (*Iijima-Ando et al., 2012*). Although there was a tendency for increased ubiquitin-positive puncta in *Miro* knockdown brains, the difference was not significant (*Figure 1B*, p>0.05 between control RNAi and *Miro* RNAi). These data suggest that the depletion of axonal mitochondria induced by *milton* knockdown leads to the accumulation of ubiquitinated proteins before neurodegeneration occurs.

It has been reported that ubiquitinated proteins accumulate with aging (*Tonoki et al., 2009*); thus, we analyzed the accumulation of ubiquitinated proteins in aged brains (30-day-old) with *milton* knockdown. The number of puncta of ubiquitinated proteins did not significantly differ between control and *milton* knockdown flies or between control and *Miro* knockdown flies (*Figure 1C*, p>0.05). These results suggest that depletion of axonal mitochondria may have more impact on proteostasis in young neurons than in old neurons.

We examined the ultrastructure of presynaptic terminals and cell bodies in photoreceptor neurons with *milton* knockdown by transmission electron microscopy in 27-day-old flies (*Figure 1D*). As previously reported (*Iijima-Ando et al., 2012*), the number of mitochondria in presynaptic terminals decreased in *milton* knockdown (*Figure 1E*). The swelling of presynaptic terminals, characterized by the enlargement and roundness, was not reported at 3-day-old (*Iijima-Ando et al., 2012*) but observed at this age with about 4% of total presynaptic terminals (*Figure 1F*, asterisks).

Some presynaptic terminals of *milton* knockdown neurons contained dense materials (*Figure 1F and G*, arrowheads). Dense materials are rarely found in age-matched control neurons, indicating that *milton* knockdown induces abnormal protein accumulation in the presynaptic terminals (*Figure 1G and H*). In *milton* knockdown neurons, dense materials are found in swollen presynaptic terminals more often than in presynaptic terminals without swelling, suggesting a positive correlation between the disruption of proteostasis and axonal damage (*Figure 1G*). In contrast, dense materials were not observed in cell bodies in the *milton* knockdown retina (*Figure 1H*). These results indicate that the depletion of axonal mitochondria induces protein accumulation in the axon.

### Depletion of axonal mitochondria impairs protein degradation pathways

Since abnormal proteins were accumulated in *milton* knockdown brains, we next examined if protein degradation pathways were suppressed. We analyzed autophagy via western blotting of the autophagy markers LC3 and p62 (*Klionsky et al., 2021*). During autophagy progression, LC3 is conjugated

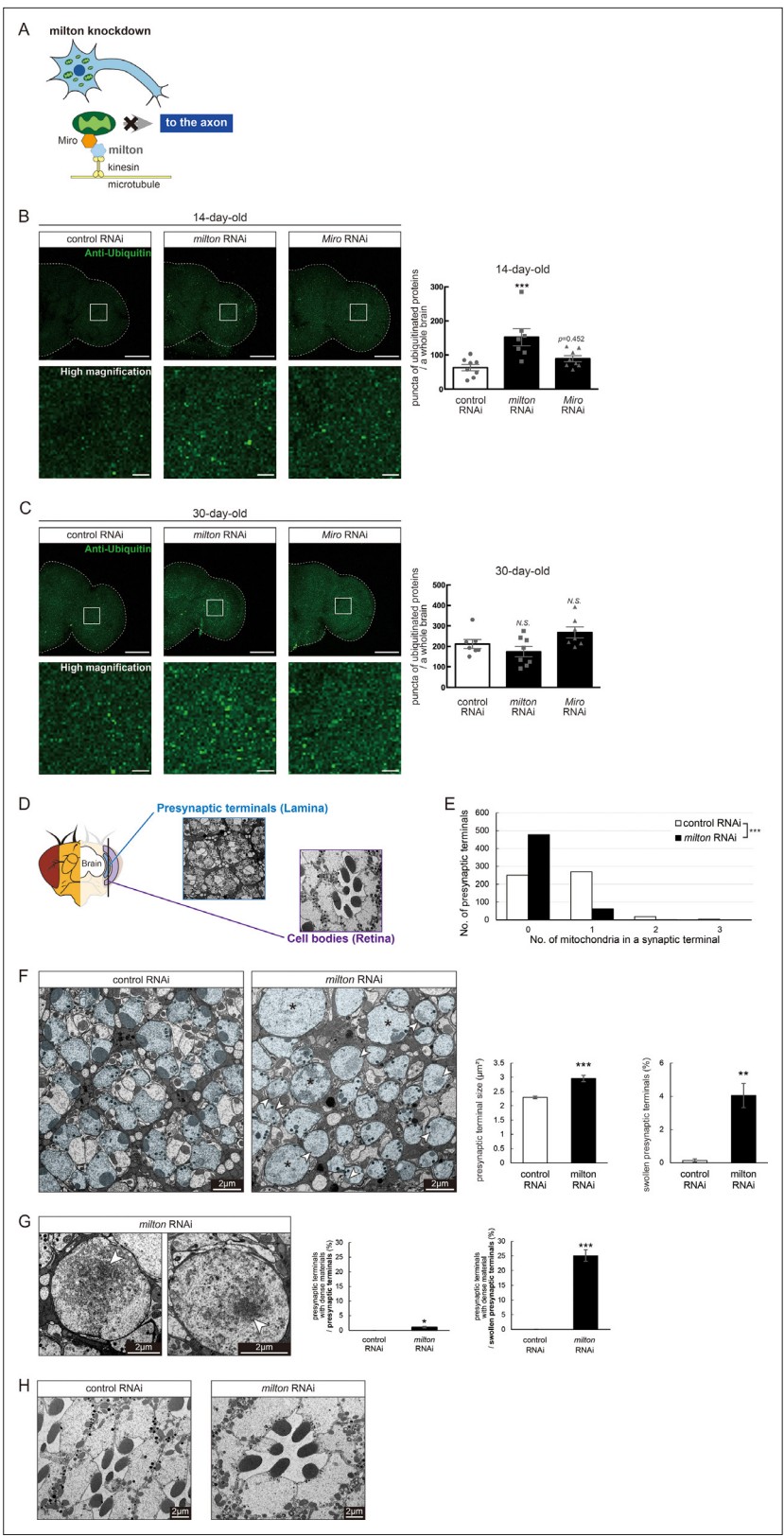

**Figure 1.** Knockdown of *milton* or *Miro* causes protein accumulation in the axon. (**A**) Schematic representation of the mitochondrial transport machinery. Knockdown of *milton*, an adapter protein for mitochondrial transport, depletes mitochondria in the axon. (**B, C**) Ubiquitinated proteins in brains with neuronal knockdown of *milton* or *Miro*. Brains dissected at 14-day-old (**B**) or 30-day-old (**C**) were immunostained with an antibody against ubiquitin.

*Figure 1 continued on next page*

*Figure 1 continued*

Firefly *luciferase* RNAi was used as a control. Representative images (left) and quantitation of the number of ubiquitin-positive puncta (right) are shown. Scale bars of hemibrains, 100 μm, Scale bars of high magnifications, 10 μm. Means ± SE, n=8. *N.S.*, p>0.05; ***p<0.005 (one-way analysis of variance (ANOVA) followed by Tukey's honestly significant difference (HSD) *post hoc* test). (**D**) Cross-sections in the lamina and in the retina were used to analyze the ultrastructure of synapses and cell bodies, respectively. *milton* RNAi was expressed in the retina and neurons via a combination of GAL4 drivers, a pan-retinal gmr-GAL4 and pan-neuronal elav-GAL4. (**E**) Quantitation of the number of mitochondria in a presynaptic terminal from transmission electron micrographs. 180 presynaptic terminals from cross-sections of the lamina from three brains were analyzed. ***p<0.005 (Chi-square test). (**F, G**) Presynaptic terminals of photoreceptor neurons of control and *milton* knockdown flies. Photoreceptor neurons are highlighted in blue. Swollen presynaptic terminals (asterisks in **F**), characterized by the enlargement and higher circularity, were found more frequently in *milton* knockdown neurons. Arrowheads indicate presynaptic terminals with dense materials. Scale bars, 2 μm. Representative images (Left) and quantitation (Right) are shown. 918–1118 from three heads were quantified for the percentage of swollen presynaptic terminals, and 180 presynaptic terminals from three heads were quantified for the size of presynaptic terminals. Mean ± SE, **p<0.01, ***p<0.005 (Student's *t*-test). (**G**) Dense materials (arrowheads in **G**) in the presynaptic terminals of *milton* knockdown neurons. Scale bars, 2 μm. The ratio of presynaptic terminals containing dense materials was quantified from 918 to 1118 presynaptic terminals from three heads. Mean ± SE, ***p<0.005 (Student's *t*-test). (**H**) Cell bodies of photoreceptor neurons of control and *milton* knockdown flies. Scale bars, 2 μm. Flies were 27-day-old.

The online version of this article includes the following figure supplement(s) for figure 1:

**Figure supplement 1.** Ubiquitinated proteins in brains with neuronal knockdown of *milton* at 1-day-old.

with phosphatidylethanolamine to form LC3-II, which localizes to isolation membranes and autophagosomes. LC3-I accumulation occurs when autophagosome formation is impaired, and LC3-II accumulation is associated with lysosomal defects (***Klionsky et al., 2021***; ***Bartlett et al., 2011***). p62 is an autophagy substrate, and its accumulation suggests autophagic defects (***Klionsky et al., 2021***; ***Bartlett et al., 2011***). We found that *milton* knockdown increased LC3-I, and the LC3-II/LC3-I ratio was lower in *milton* knockdown flies than in control flies at 14-day-old (***Figure 2A***). We also analyzed p62 levels in head lysates sequentially extracted using detergents with different stringencies (1% Triton X-100 and 2% SDS). Western blotting revealed that p62 levels were increased in the brains of 14-day-old *milton* knockdown flies (***Figure 2B***). The increase in the p62 level was significant in the Triton X-100-soluble fraction but not in the SDS-soluble fraction (***Figure 2B***), suggesting that depletion of axonal mitochondria impairs the degradation of less-aggregated proteins. Proteasome activity was also significantly decreased in brains with neuronal knockdown of *milton* (***Figure 2C***, p<0.005).

At 30 days old, LC3-I was still higher, and the LC3-II/LC3-I ratio was lower, in *milton* knockdown compared to the control (***Figure 2D***). At this age, *milton* knockdown increased p62 significantly in 1% Triton X-100 fraction and 2% SDS fraction (***Figure 2E***). Proteasome activities were also decreased in *milton* knockdown flies at 30-day-old (***Figure 2F***). These results indicate that depletion of axonal mitochondria impairs protein degradation pathways.

## ATP deprivation does not impair autophagy

*milton* knockdown downregulates ATP in the axon (***Oka et al., 2021***). To examine whether the disruption of protein degradation pathways by *milton* knockdown is due to ATP deprivation, we investigated the effects of knocking down phosphofructokinase (*Pfk*), a rate-limiting enzyme in glycolysis, on protein degradation pathways. Neuronal knockdown of *Pfk* was reported to lower ATP levels in brain neurons (***Oka et al., 2021***). *Pfk* knockdown and *milton* knockdown decreased ATP to similar levels (***Figure 3A–C***). However, in contrast with *milton* knockdown, *Pfk* knockdown did not affect the levels of LC3-I, LC3-II, or the LC3-II/LC3-I ratio (***Figure 3D***). *Pfk* knockdown decreased p62 level (***Figure 3E***), suggesting that autophagy is promoted. On the other hand, proteasome activity was decreased by *Pfk* knockdown (***Figure 3F***). These results suggest that the downregulation of axonal ATP upon depletion of axonal mitochondria decreases proteasome activity, but not autophagy.

## Proteome analysis suggests that depletion of axonal mitochondria causes disruption of autophagy and premature aging

To identify the pathways that mediate the decrease in autophagy in *milton* knockdown brains, we performed proteome analysis to systematically detect differentially expressed proteins upon neuronal

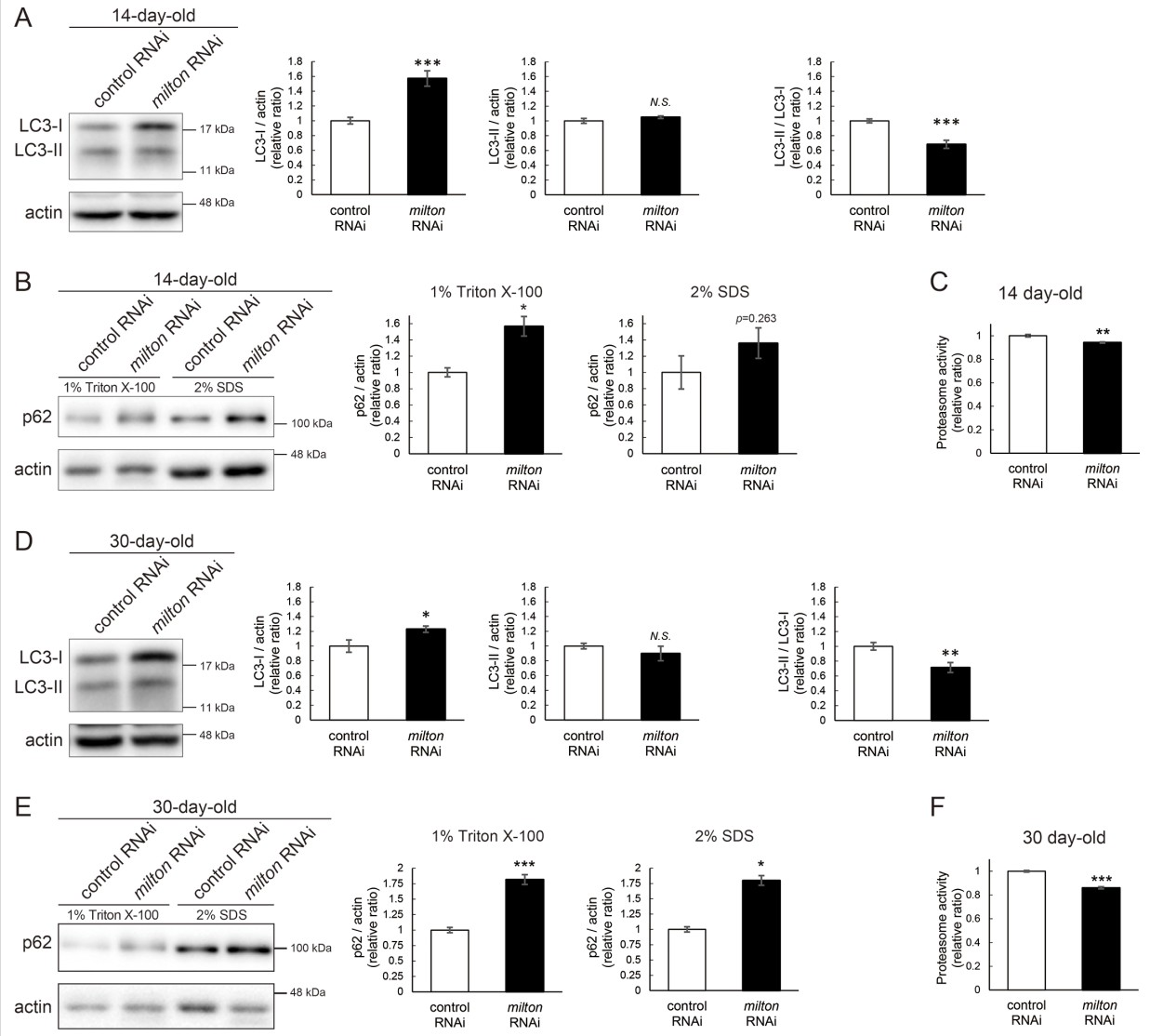

**Figure 2.** *milton* knockdown impairs protein degradation pathways. (**A, B**) Western blotting of head extracts of control and *milton* knockdown flies with antibodies against LC3 (**A**) and Ref2P, the fly homolog of mammalian p62 (**B**). For the analyses of p62 levels, heads were extracted with 1% Triton X-100 or 2% SDS (**B**). Flies were 14-day-old. Representative blots (left) and quantitation (right) are shown. Actin was used as a loading control. Means ± SE, n=6 (LC3), n=3 (p62). (**C**) Proteasome activity in head extracts of control and *milton* knockdown flies was measured by hydrolysis of Suc-LLVY-AMC at 14-day-old. Means ± SE, n=3. (**D, E**) Western blotting of head extracts of 30-day-old control and *milton* knockdown flies. Blotting was performed with anti-LC3 (**D**) and anti-p62 (**E**) antibodies. Representative blots (left) and quantitation (right) are shown. Actin was used as a loading control. Means ± SE, n=6 (LC3), n=3 (p62). (**F**) Proteasome activity in head extracts of 30-day-old control and *milton* knockdown flies. Means ± SE, n=3. *N.S.*, p>0.05; *p<0.05; **p<0.01; ***p<0.005 (Student's *t*-test).

The online version of this article includes the following source data for figure 2:

**Source data 1.** PDF file containing original western blots for *Figure 2*, indicating the relevant bands.

**Source data 2.** Original files for western blot analysis displayed in *Figure 2*.

knockdown of *milton*. We analyzed flies at 7- and 21-day-old, the age before autophagic defects are detected and the age just before the onset of neurodegeneration, respectively (*Figure 4A*). 1039 proteins were detected by liquid chromatography-tandem mass spectrometry (LC-MS/MS). Expression of 36 proteins was significantly increased (22 proteins) or decreased (14 proteins) by *milton* knockdown at 7-day-old (*Figure 4B*, *Table 1* and *Supplementary file 1*). At 21 days old, the expression of 41 proteins (31 upregulated and ten downregulated proteins) was significantly altered in *milton* knockdown flies compared with control flies (*Figure 4C*, *Table 1* and *Supplementary file 1*).

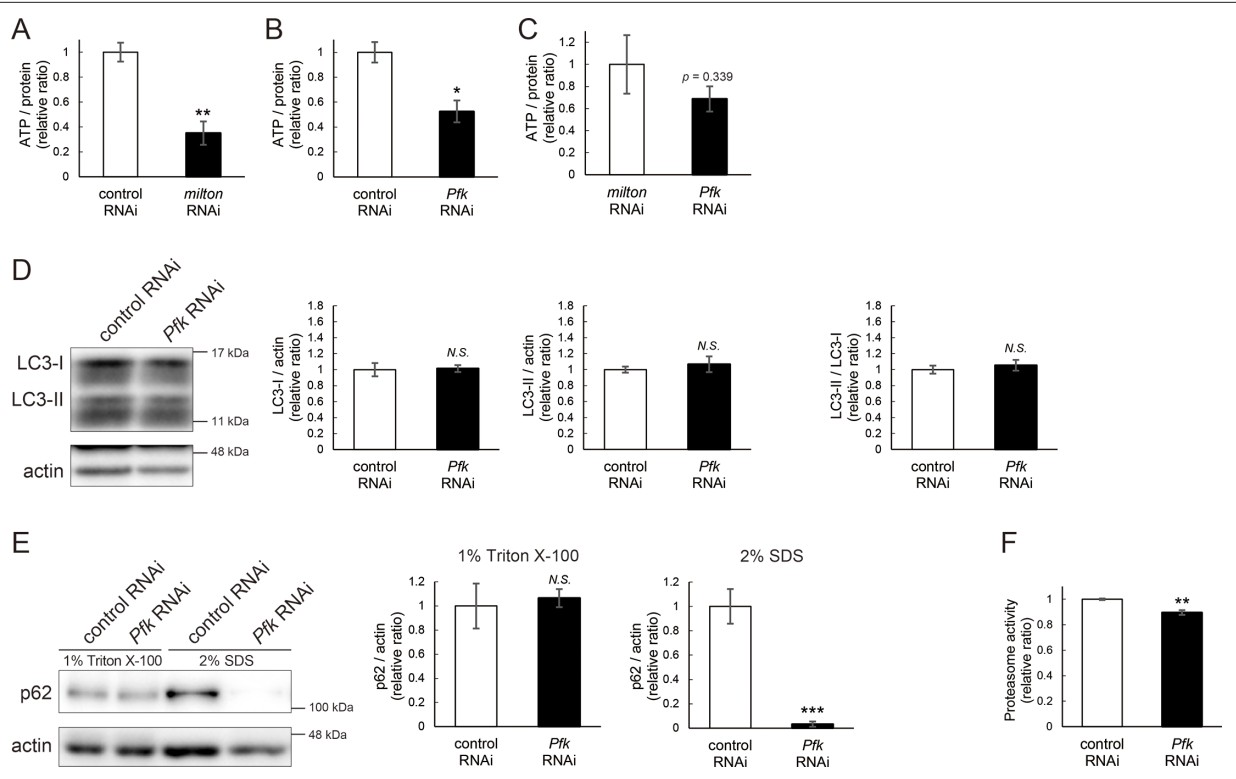

**Figure 3.** ATP deprivation does not impair autophagy. (**A–C**) ATP levels in brain extracts of control and *milton* knockdown flies (**A**) and control and *Pfk* knockdown flies (**B**) and comparison of the effects of *milton* knockdown and *Pfk* knockdown on ATP levels (**C**). Flies were 14-day-old. Means ± SE, n=3. (**D, E**) Western blotting of head extracts of flies with neuronal expression of control or *Pfk* RNAi. Blotting was performed with anti-LC3 (**D**) and anti-p62 (**E**) antibodies. For analyses of p62 levels, heads were extracted with 1% Triton X-100 or 2% SDS. Representative blots (left) and quantitation (right) are shown. Actin was used as a loading control. Means ± SE, n=6 (LC3), n=3 (p62). (**F**) Proteasome activity in head lysates of flies with neuronal expression of control or *Pfk* RNAi was measured by hydrolysis of Suc-LLVY-AMC. Means ± SE, n=3. *N.S.*, p>0.05; *p<0.05; **p<0.01; ***p<0.005 (Student's *t*-test). Flies were 14 days old.

The online version of this article includes the following source data for figure 3:

**Source data 1.** PDF file containing original western blots for *Figure 3* indicating the relevant bands.

**Source data 2.** Original files for western blot analysis displayed in *Figure 3*.

The 'Interaction search' algorithm using KeyMolnet showed that proteins whose expression was significantly altered in the brains of *milton* knockdown flies at both 7- and 21-day-old were closely associated with the autophagic pathway (*Table 2*). Proteins involved in pathways characteristics of aging, such as the immune response (transcriptional regulation by STAT), cancer (transcriptional regulation by SMAD, transcriptional regulation by myc), longevity (transcriptional regulation by FOXO, Sirtuin signaling pathway), and stress responses (HSP90 signaling pathway, MAPK signaling pathway; *Zia et al., 2021*; *Haigis and Yankner, 2010*), were enriched in the proteome profiles of *milton* knockdown flies compared with those of control flies at 7-day-old (*Table 2*), suggesting that depletion of axonal mitochondria accelerates aging in the brain.

## Depletion of axonal mitochondria upregulates eIF2β and decreases phosphorylation of eIF2α

Differentially expressed proteins at 7-day-old flies may reflect alterations that are causal for autophagic defects. We noticed that the expression level of eIF2β was 2.465-fold higher in the brains of *milton* knockdown flies than in those of control flies (*Figure 4B and D*). Upregulation of eIF2β in the brains of *milton* knockdown flies was confirmed by western blotting. *milton* knockdown increased eIF2β protein levels more than twice (*Figure 4E*), but did not change the level of *eIF2β* mRNA (*Figure 4F*).

We also investigated age-dependent changes in eIF2β by western blotting of control flies at 7-, 21-, 35-, 49-, and 63-day-old. eIF2β levels increased during aging until 49-day-old (*Figure 4G*). These

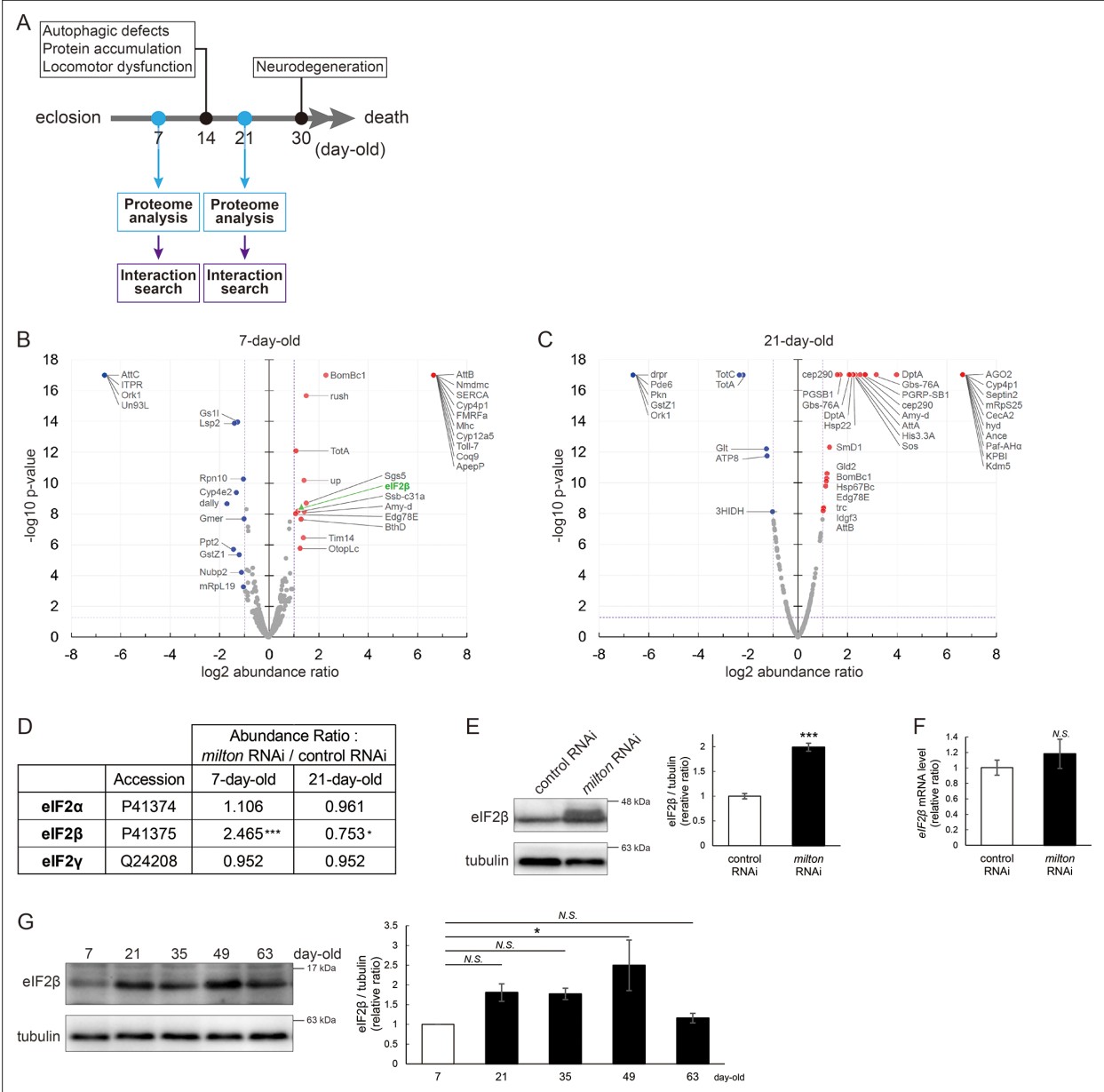

**Figure 4.** *milton* knockdown upregulates eIF2β in young flies. (**A**) Timing of proteome analysis and phenotypes observed in *milton* knockdown flies. (**B**) and (**C**) Volcano plots of the $log_2$ abundance ratio (x-axis) against the $-log_{10}$ p-value (y-axis) of proteins at 7 days old (**B**) and 21 days old (**C**). (**D**) eIF2 subunit protein levels from proteome analysis of *milton* knockdown flies compared to those of control flies. (**E**) Western blotting of head extracts of flies expressing control or *milton* RNAi in neurons with an anti-eIF2β antibody. Flies were 14-day-old. Representative blots (left) and quantitation (right) are shown. Tubulin was used as a loading control. Means ± SE, n=6. (**F**) *eIF2β* mRNA levels quantified by qRT-PCR. Means ± SE, n=4. (**G**) Western blotting of head extracts of wild-type flies with an anti-eIF2β antibody. Flies were 7-, 21-, 35-, 49-, and 63-day-old. Representative blots (left) and quantitation (right) are shown. Tubulin was used as a loading control. Means ± SE, n=3, *p<0.05 (one-way analysis of variance (ANOVA) followed by Dunnett's multiple comparison test).

The online version of this article includes the following source data for figure 4:

**Source data 1.** PDF file containing original western blots for *Figure 4*, indicating the relevant bands.

**Source data 2.** Original files for western blot analysis displayed in *Figure 4*.

results suggest that upregulation of eIF2β in *milton* knockdown fly brain reflects early an onset of age-dependent increase of eIF2β levels.

eIF2β is a subunit of the eukaryotic initiation factor 2 (eIF2) complex, which is critical for translation initiation and the integrated stress response (ISR; **Kimball, 1999**). eIF2 is a heterotrimer of α, β, and

**Table 1.** Differentially expressed proteins in milton RNAi fly brains compared to control at 7 day-old detected by proteome analysis.

**7-day-old**

| Accession* | Name | Abundance ratio:(7 days, milton KD)/ (7 days, control) | Abundance ratio p-value: (7 days, milton KD) / (7 days, control) |
|---|---|---|---|
| Q9V751 | Attacin-B | 100 | 1E-17 |
| Q04448 | Bifunctional methylenetetrahydrofolate dehydrogenase/cyclohydrolase, mitochondrial | 100 | 1E-17 |
| P22700 | Calcium-transporting ATPase sarcoplasmic/endoplasmic reticulum type | 100 | 1E-17 |
| Q9V558 | Cytochrome P450 4p1 | 100 | 1E-17 |
| P10552 | FMRFamide-related peptides | 100 | 1E-17 |
| P05661-19 | Isoform F of Myosin heavy chain, muscle | 100 | 1E-17 |
| Q9VE01 | Probable cytochrome P450 12a5, mitochondrial | 100 | 1E-17 |
| Q7KIN0 | Toll-like receptor 7 | 100 | 1E-17 |
| Q8MKN0 | Ubiquinone biosynthesis protein COQ9, mitochondrial | 100 | 1E-17 |
| Q9VJG0 | Xaa-Pro aminopeptidase ApepP | 100 | 1E-17 |
| Q9V8F5 | Bomanin Bicipital 1 | 4.908 | 1E-17 |
| P07701 | Salivary glue protein Sgs-5 | 2.843 | 1.99252E-09 |
| O76902 | Pleckstrin homology domain-containing family F member 1 homolog | 2.836 | 2.22045E-16 |
| P81641 | Alpha-amylase B | 2.684 | 7.44847E-09 |
| P19351-4 | Isoform 4 of Troponin T, skeletal muscle | 2.66 | 6.65563E-11 |
| Q9VTJ8 | Mitochondrial import inner membrane translocase subunit TIM14 | 2.61 | 3.54205E-07 |
| P41375 | Eukaryotic translation initiation factor 2 subunit 2 | 2.465 | 3.38486E-09 |
| Q9VYB0 | Selenoprotein BthD | 2.462 | 2.25579E-08 |
| B7Z0W9 | Proton channel OtopLc | 2.382 | 1.71741E-06 |
| Q9VLR5 | RNA polymerase II transcriptional coactivator | 2.245 | 6.70245E-09 |
| Q8IN44 | Protein Turandot A | 2.127 | 8.38662E-13 |
| P27779 | Pupal cuticle protein Edg-78E | 2.113 | 1.00215E-08 |
| Q9W1X8 | Probable GDP-L-fucose synthase | 0.496 | 2.12601E-08 |
| P55035 | 26 S proteasome non-ATPase regulatory subunit 4 | 0.487 | 5.50158E-11 |
| Q9VHN6 | 39 S ribosomal protein L19, mitochondrial | 0.487 | 0.000551337 |
| Q9VPD2 | Cytosolic Fe-S cluster assembly factor NUBP2 homolog | 0.46 | 6.34514E-05 |
| Q9VHD3 | Probable maleylacetoacetate isomerase 1 | 0.432 | 4.50956E-06 |
| Q94529 | Probable pseudouridine-5'-phosphatase | 0.416 | 1.08802E-14 |
| Q27606 | Cytochrome P450 4e2 | 0.398 | 4.13128E-10 |
| Q24388 | Larval serum protein 2 | 0.378 | 1.33227E-14 |
| Q9VKH6 | Lysosomal thioesterase PPT2 homolog | 0.369 | 2.05225E-06 |
| Q24114 | Division abnormally delayed protein | 0.307 | 2.24406E-09 |
| Q95NH6 | Attacin-C | 0.01 | 1E-17 |
| P29993 | Inositol 1,4,5-trisphosphate receptor | 0.01 | 1E-17 |
| Q94526 | Open rectifier potassium channel protein 1 | 0.01 | 1E-17 |

*Table 1 continued on next page*

*Table 1 continued*

**7-day-old**

| Accession* | Name | Abundance ratio:(7 days, milton KD)/ (7 days, control) | Abundance ratio p-value: (7 days, milton KD) / (7 days, control) |
|---|---|---|---|
| Q9Y115 | UNC93-like protein | 0.01 | 1E-17 |

**21-day-old**

| Accession* | Name | Abundance ratio:(21 days, milton KD) /(21 days, control) | Abundance ratio p-value:(21 days, milton KD) /(21 days, control) |
|---|---|---|---|
| Q10714 | Angiotensin-converting enzyme | 100 | 1E-17 |
| C0HKQ8 | Cecropin-A2 | 100 | 1E-17 |
| Q9V558 | Cytochrome P450 4p1 | 100 | 1E-17 |
| P51592 | E3 ubiquitin-protein ligase hyd | 100 | 1E-17 |
| Q9VMJ7 | Lysine-specific demethylase lid | 100 | 1E-17 |
| Q9VXP4 | Platelet-activating factor acetylhydrolase IB subunit beta homolog | 100 | 1E-17 |
| Q9VY28 | Probable 28 S ribosomal protein S25, mitochondrial | 100 | 1E-17 |
| Q9W391 | Probable phosphorylase b kinase regulatory subunit alpha | 100 | 1E-17 |
| Q9VUQ5 | Protein argonaute-2 | 100 | 1E-17 |
| P54359 | Septin-2 | 100 | 1E-17 |
| P24492 | Diptericin A | 15.716 | 1E-17 |
| Q9VVY3 | Glycogen-binding subunit 76 A | 8.986 | 1E-17 |
| Q70PY2 | Peptidoglycan-recognition protein SB1 | 6.669 | 1E-17 |
| Q9W0M1 | Centrosomal protein cep290 | 6.526 | 1E-17 |
| P81641 | Alpha-amylase B | 5.722 | 1E-17 |
| P45884 | Attacin-A | 4.997 | 1E-17 |
| C0HL66 | Histone H3.3A | 4.778 | 1E-17 |
| P26675 | Protein son of sevenless | 4.696 | 1E-17 |
| P02515 | Heat shock protein 22 | 4.69 | 1E-17 |
| Q95NH6 | Attacin-C | 4.35 | 1E-17 |
| P17971-1 | Isoform A of Potassium voltage-gated channel protein Shal | 4.195 | 1E-17 |
| Q7K1U0 | Activity-regulated cytoskeleton associated protein 1 | 3.271 | 1E-17 |
| P14199 | Protein ref(2)P | 3.014 | 1E-17 |
| Q9VU02 | Probable small nuclear ribonucleoprotein Sm D1 | 2.43 | 4.91607E-13 |
| Q9VD44 | Poly(A) RNA polymerase gld-2 homolog A | 2.268 | 2.6084E-11 |
| Q9V8F5 | Bomanin Bicipital 1 | 2.24 | 5.16671E-11 |
| P22979 | Heat shock protein 67B3 | 2.223 | 7.90048E-11 |
| P27779 | Pupal cuticle protein Edg-78E | 2.192 | 1.65944E-10 |
| Q9NBK5 | Serine/threonine-protein kinase tricornered | 2.059 | 4.15071E-09 |
| Q8MLZ7 | Chitinase-like protein Idgf3 | 2.055 | 4.61182E-09 |
| Q9V751 | Attacin-B | 2.038 | 6.79416E-09 |

Table 1 continued

**21-day-old**

| Accession* | Name | Abundance ratio:(21 days, milton KD) /(21 days, control) | Abundance ratio p-value:(21 days, milton KD) /(21 days, control) |
|---|---|---|---|
| Q9V8M5 | Probable 3-hydroxyisobutyrate dehydrogenase, mitochondrial | 0.492 | 7.42952E-09 |
| P84345 | ATP synthase protein 8 | 0.421 | 1.809E-12 |
| P33438 | Glutactin | 0.414 | 6.13731E-13 |
| Q8IN44 | Protein Turandot A | 0.218 | 1E-17 |
| Q8IN43 | Protein Turandot C | 0.195 | 1E-17 |
| Q9VFI9 | cGMP-specific 3',5'-cyclic phosphodiesterase | 0.01 | 1E-17 |
| Q94526 | Open rectifier potassium channel protein 1 | 0.01 | 1E-17 |
| Q9VHD3 | Probable maleylacetoacetate isomerase 1 | 0.01 | 1E-17 |
| Q9W0A0 | Protein draper | 0.01 | 1E-17 |
| A1Z7T0 | Serine/threonine-protein kinase N | 0.01 | 1E-17 |

*UniProt accession number.

γ subunits, and eIF2α is phosphorylated during the ISR (**Pakos-Zebrucka et al., 2016**). As for the other subunits of the eIF2 complex, proteome analysis did not detect a significant difference in the protein levels of eIF2α and eIF2γ between *milton* knockdown and control flies at 7- and 21-day-old (**Figure 4D**). Western blotting of brain lysates showed that *milton* knockdown reduced eIF2α levels (**Figure 5A**), while p-eIF2α levels were not significantly affected (**Figure 5B**).

To analyze local changes of eIF2α and p-eIF2α, we carried out immunostaining. We focused on the mushroom body, where axons, dendrites, and cell bodies can be easily identified (**Figure 5C**). Both eIF2α and p-eIF2α were downregulated in the cell body (Kenyon cells) and dendritic (Calyxes) regions of the brains of *milton* knockdown flies (**Figure 5D**). In axons (lobe tips), *milton* knockdown did not affect eIF2α (**Figure 5E**, p=0.271) but significantly downregulated p-eIF2α (**Figure 5E**). The ratio of p-eIF2α to eIF2α was lower in the axon but not in the soma or dendritic region. These results suggest that axonal distribution of mitochondria regulates the level of overall eIF2α protein and local p-eIF2α.

## Depletion of axonal mitochondria suppressed global translation

Phosphorylation of eIF2α induces conformational changes in the eIF2 complex and inhibits global translation (**Wek, 2018**). To analyze the effects of *milton* knockdown on translation, we performed polysome gradient centrifugation to examine the level of ribosome binding to mRNA. Since p-eIF2α was downregulated, we hypothesized that *milton* knockdown would enhance translation. However, unexpectedly, we found that *milton* knockdown significantly reduced the level of mRNAs associated with polysomes (**Figure 6A and B**). We also compared the level of translation between the brains of control and *milton* knockdown flies by assessing the incorporation of puromycin (**Figure 6C**). Puromycin incorporation was lower in the brains of *milton* knockdown flies than in those of control flies, while it was not statistically significant (**Figure 6C**, indicated by a bracket). These data suggest that the depletion of axonal mitochondria suppresses global translation.

## eIF2β upregulation reduces the level of p-eIF2α, impairs autophagy, and decreases locomotor function

We were motivated to ask if eIF2β upregulation mediates autophagic defects caused by *milton* knockdown. If so, neuronal overexpression of *eIF2β* would also induce autophagy impairment. Neuronal overexpression of *eIF2β* increased LC3-II, while the LC3-II/LC3-I ratio was not significantly different (**Figure 7A and B**). Overexpression of *eIF2β* significantly increased the p62 level in the Triton X-100-soluble fraction (**Figure 7C**, fourfold vs. control, p<0.005 [1% Triton X-100]) but not in the SDS-soluble fraction (**Figure 7C**, twofold vs. control, p=0.062 [2% SDS]), as observed in brains of *milton* knockdown

**Table 2.** Molecule networks based on "Interaction search" of KeyMolnet.

**7-day-old**

| Rank | Name | Score | Score (p)* | Score (v)† | Score (c)‡ |
|---|---|---|---|---|---|
| 1 | Autophagy-related protein signaling pathway | 50.394 | 6.76E-16 | 0.159 | 0.11 |
| 2 | Calcium signaling pathway | 47.583 | 4.75E-15 | 0.146 | 0.117 |
| 3 | Transcriptional regulation by SMAD | 44.012 | 5.64E-14 | 0.146 | 0.095 |
| 4 | GABA signaling pathway | 40.706 | 5.58E-13 | 0.122 | 0.123 |
| 5 | estrogen signaling pathway | 37.507 | 5.12E-12 | 0.11 | 0.13 |
| 6 | Sirtuin signaling pathway | 36.87 | 7.96E-12 | 0.122 | 0.095 |
| 7 | Transcriptional regulation by AP-1 | 34.874 | 3.18E-11 | 0.11 | 0.107 |
| 8 | Arrestin signaling pathway | 32.84 | 1.30E-10 | 0.11 | 0.092 |
| 9 | G protein (Gq/11) signaling pathway | 30.889 | 5.03E-10 | 0.085 | 0.149 |
| 10 | Kainate receptor signaling pathway | 30.049 | 9.00E-10 | 0.073 | 0.214 |
| 11 | Transcriptional regulation by C/EBP | 29.5 | 1.32E-09 | 0.098 | 0.093 |
| 12 | Calpain signaling pathway | 28.597 | 2.46E-09 | 0.11 | 0.066 |
| 13 | Phospholipase D signaling pathway | 28.344 | 2.94E-09 | 0.098 | 0.084 |
| 14 | HSP90 signaling pathway | 27.188 | 6.54E-09 | 0.085 | 0.104 |
| 14 | CYP family | 27.188 | 6.54E-09 | 0.085 | 0.104 |
| 16 | Kir3 channel signaling pathway | 26.495 | 1.06E-08 | 0.061 | 0.25 |
| 17 | Estrogen biosynthesis | 26.107 | 1.38E-08 | 0.061 | 0.238 |
| 18 | CaSR signaling pathway | 25.39 | 2.27E-08 | 0.061 | 0.217 |
| 19 | PI3K signaling pathway | 24.927 | 3.14E-08 | 0.073 | 0.122 |
| 20 | PAF receptor signaling pathway | 24.555 | 4.06E-08 | 0.049 | 0.4 |
| 21 | Transcriptional regulation by PPARa | 24.398 | 4.52E-08 | 0.073 | 0.115 |
| 21 | BTK signaling pathway | 24.398 | 4.52E-08 | 0.073 | 0.115 |
| 23 | Transcriptional regulation by STAT | 24.076 | 5.66E-08 | 0.085 | 0.077 |
| 24 | G protein (Gi/o) signaling pathway | 24.063 | 5.71E-08 | 0.073 | 0.111 |
| 25 | PARP signaling pathway | 23.742 | 7.13E-08 | 0.073 | 0.107 |
| 25 | mGluR signaling pathway | 23.742 | 7.13E-08 | 0.073 | 0.107 |
| 27 | Free fatty acid signaling pathway | 23.433 | 8.83E-08 | 0.073 | 0.103 |
| 28 | Kir channel signaling pathway | 23.338 | 9.43E-08 | 0.061 | 0.167 |
| 29 | Oxytocin signaling pathway | 23.327 | 9.51E-08 | 0.049 | 0.333 |
| 30 | Transcriptional regulation by MEF2 | 22.99 | 1.20E-07 | 0.073 | 0.098 |
| 31 | S100 family signaling pathway | 22.434 | 1.77E-07 | 0.073 | 0.092 |
| 32 | Transcriptional regulation by FOXO | 22.301 | 1.94E-07 | 0.073 | 0.091 |
| 33 | P2Y signaling pathway | 22.172 | 2.12E-07 | 0.061 | 0.143 |
| 34 | Transcriptional regulation by SRF | 21.174 | 4.23E-07 | 0.061 | 0.125 |
| 34 | ATF4/ATF6/IRE1 signaling pathway | 21.174 | 4.23E-07 | 0.061 | 0.125 |
| 36 | Chemerin signaling pathway | 21.082 | 4.50E-07 | 0.049 | 0.235 |
| 36 | Vasopressin signaling pathway | 21.082 | 4.50E-07 | 0.049 | 0.235 |
| 38 | Serotonin signaling pathway | 20.854 | 5.28E-07 | 0.073 | 0.077 |

*Table 2 continued on next page*

*Table 2 continued*

**7-day-old**

| Rank | Name | Score | Score (p)* | Score (v)† | Score (c)‡ |
|---|---|---|---|---|---|
| 39 | Transcriptional regulation by HIF | 20.834 | 5.35E-07 | 0.098 | 0.043 |
| 40 | Leukotriene receptor signaling pathway | 20.724 | 5.78E-07 | 0.049 | 0.222 |
| 40 | CART signaling pathway | 20.724 | 5.78E-07 | 0.049 | 0.222 |
| 42 | MAPK signaling pathway | 20.693 | 5.90E-07 | 0.085 | 0.055 |
| 43 | Transcriptional regulation by RB/E2F | 20.543 | 6.55E-07 | 0.098 | 0.042 |
| 44 | NAD metabolism | 20.468 | 6.89E-07 | 0.061 | 0.114 |
| 45 | ERK signaling pathway | 20.425 | 7.11E-07 | 0.073 | 0.073 |
| 46 | Adenylyl Cyclase signaling pathway | 20.303 | 7.73E-07 | 0.061 | 0.111 |
| 47 | Bile acid signaling pathway | 20.141 | 8.65E-07 | 0.061 | 0.109 |

**21-day-old**

| Rank | Name | Score | Score (p)* | Score (v)† | Score (c)‡ |
|---|---|---|---|---|---|
| 1 | Histone demethylation | 84.198 | 4.51E-26 | 0.102 | 0.425 |
| 2 | CDK inhibitor signaling pathway | 56.497 | 9.83E-18 | 0.078 | 0.295 |
| 3 | Transcriptional regulation by RB/E2F | 46.598 | 9.39E-15 | 0.108 | 0.095 |
| 4 | Mst(Hippo) signaling pathway | 46.343 | 1.12E-14 | 0.09 | 0.133 |
| 5 | Transcriptional regulation by androgen receptor | 46.078 | 1.35E-14 | 0.078 | 0.178 |
| 6 | p160 SRC signaling pathway | 45.809 | 1.62E-14 | 0.078 | 0.176 |
| 7 | Transcriptional regulation by SMAD | 43.961 | 5.84E-14 | 0.09 | 0.119 |
| 8 | Autophagy-related protein signaling pathway | 41.063 | 4.35E-13 | 0.084 | 0.119 |
| 9 | Transcriptional regulation by HIF | 39.527 | 1.26E-12 | 0.096 | 0.086 |
| 10 | Nucleophosmin signaling pathway | 38.417 | 2.72E-12 | 0.054 | 0.273 |
| 11 | HSP90 signaling pathway | 37.887 | 3.93E-12 | 0.066 | 0.164 |
| 12 | PAF metabolism | 37.562 | 4.93E-12 | 0.042 | 0.5 |
| 13 | Transcriptional regulation by STAT | 37.276 | 6.01E-12 | 0.072 | 0.132 |
| 14 | Bcl-2 family signaling pathway | 36.157 | 1.31E-11 | 0.072 | 0.124 |
| 15 | Sirtuin signaling pathway | 34.782 | 3.39E-11 | 0.072 | 0.114 |
| 16 | Transcriptional regulation by C/EBP | 33.819 | 6.60E-11 | 0.066 | 0.128 |
| 17 | PIN1 signaling pathway | 33.172 | 1.03E-10 | 0.06 | 0.149 |
| 18 | RSK signaling pathway | 30.566 | 6.29E-10 | 0.06 | 0.125 |
| 19 | Transcriptional regulation by High mobility group protein | 29.873 | 1.02E-09 | 0.054 | 0.148 |
| 20 | BET family signaling pathway | 29.656 | 1.18E-09 | 0.054 | 0.145 |
| 21 | Transcriptional regulation by Myc | 28.838 | 2.08E-09 | 0.066 | 0.093 |
| 22 | Transcriptional regulation by FOXO | 28.827 | 2.10E-09 | 0.054 | 0.136 |
| 23 | PSD-95 family signaling pathway | 26.154 | 1.34E-08 | 0.048 | 0.14 |
| 24 | AKT signaling pathway | 25.169 | 2.65E-08 | 0.048 | 0.129 |
| 25 | Arginine methylation | 24.799 | 3.43E-08 | 0.048 | 0.125 |
| 26 | gp130 signaling pathway | 24.25 | 5.01E-08 | 0.054 | 0.096 |
| 27 | Transcriptional regulation by CREB | 23.858 | 6.58E-08 | 0.066 | 0.067 |

*Table 2 continued on next page*

*Table 2 continued*

**21-day-old**

| Rank | Name | Score | Score (p)[*] | Score (v)[†] | Score (c)[‡] |
|------|------|-------|----------|----------|----------|
| 28 | Gene regulation by microRNAs (metastasis) | 23.852 | 6.60E-08 | 0.054 | 0.093 |
| 29 | HDAC signaling pathway | 23.536 | 8.22E-08 | 0.036 | 0.207 |
| 30 | Calpain signaling pathway | 23.092 | 1.12E-07 | 0.06 | 0.074 |
| 31 | Transcriptional regulation by IRF | 22.738 | 1.43E-07 | 0.054 | 0.085 |
| 32 | 2-Oxoglutarate signaling pathway | 22.673 | 1.50E-07 | 0.048 | 0.104 |
| 32 | 14-3-3 signaling pathway | 22.673 | 1.50E-07 | 0.048 | 0.104 |
| 34 | Transcriptional regulation by POU domain factor | 22.601 | 1.57E-07 | 0.06 | 0.071 |
| 35 | Transcriptional regulation by BLIMP-1 | 22.474 | 1.72E-07 | 0.042 | 0.132 |
| 36 | Gene regulation by microRNAs (metabolism) | 22.39 | 1.82E-07 | 0.054 | 0.083 |
| 37 | Fatty acid beta oxidation | 22.096 | 2.23E-07 | 0.042 | 0.127 |
| 38 | Transcriptional regulation by RXR | 22.08 | 2.26E-07 | 0.036 | 0.176 |
| 39 | ERK signaling pathway | 21.96 | 2.45E-07 | 0.048 | 0.098 |
| 40 | PARP signaling pathway | 21.913 | 2.53E-07 | 0.042 | 0.125 |
| 41 | Transcriptional regulation by VDR | 21.618 | 3.11E-07 | 0.054 | 0.078 |
| 42 | Transcriptional regulation by p53 | 21.168 | 4.24E-07 | 0.072 | 0.05 |
| 43 | Acetylcholine metabolism | 21.152 | 4.29E-07 | 0.024 | 0.444 |
| 44 | Gene regulation by microRNAs (embryonic stem cells) | 21.08 | 4.51E-07 | 0.036 | 0.158 |
| 45 | mTOR signaling pathway | 21.048 | 4.61E-07 | 0.042 | 0.115 |
| 46 | Gene regulation by microRNAs (cancer) | 21.04 | 4.64E-07 | 0.048 | 0.09 |
| 47 | Transcriptional regulation by Ets-1/2 | 20.724 | 5.77E-07 | 0.042 | 0.111 |
| 48 | MAPK signaling pathway | 20.411 | 7.18E-07 | 0.054 | 0.07 |
| 49 | Gene regulation by microRNAs (cell cycle) | 20.404 | 7.21E-07 | 0.036 | 0.146 |
| 50 | Transcriptional regulation by p73 | 20.259 | 7.97E-07 | 0.042 | 0.106 |

[*]Score(p) indicates p-value of the pathway.

[†]Score(v) indicates the ratio of 'Count' to total molecules associated with the loaded list.

[‡]Score(c) indicates the ratio of 'Count' to total molecules contained in the pathway.

flies (*Figure 2B*). These data suggest that neuronal overexpression of *eIF2β* accumulates autophagic substrates.

Since the *milton* knockdown reduced the p-eIF2α level (*Figure 5E*), we asked whether an increase in eIF2β affects p-eIF2α. Neuronal overexpression of *eIF2β* did not affect the eIF2α level but significantly decreased the p-eIF2α level (*Figure 7D and E*).

Depletion of axonal mitochondria causes age-dependent decline in locomotor function (*Iijima-Ando et al., 2012*). We found that neuronal overexpression of *eIF2β* also caused locomotor dysfunction (*Figure 7F*). Locomotor functions were significantly impaired in those flies at 20 days old and worsened further during aging (*Figure 7F*, compare 4-, 20-, and 30-day-old). We asked if *eIF2β* overexpression causes neurodegeneration, as depletion of axonal mitochondria in the photoreceptor neurons causes axon degeneration in an age-dependent manner (*Iijima-Ando et al., 2012*). *eIF2β* overexpression in photoreceptor neurons tends to increase neurodegeneration in aged flies, while it was not statistically significant (p>0.05, *Figure 7—figure supplement 1*).

These data indicate that an increase of eIF2β in neurons phenocopies depletion of axonal mitochondria, including suppression of autophagy and age-dependent locomotor dysfunction, and suggest that increase of eIF2β mediates these phenotypes downstream of loss of axonal mitochondria.

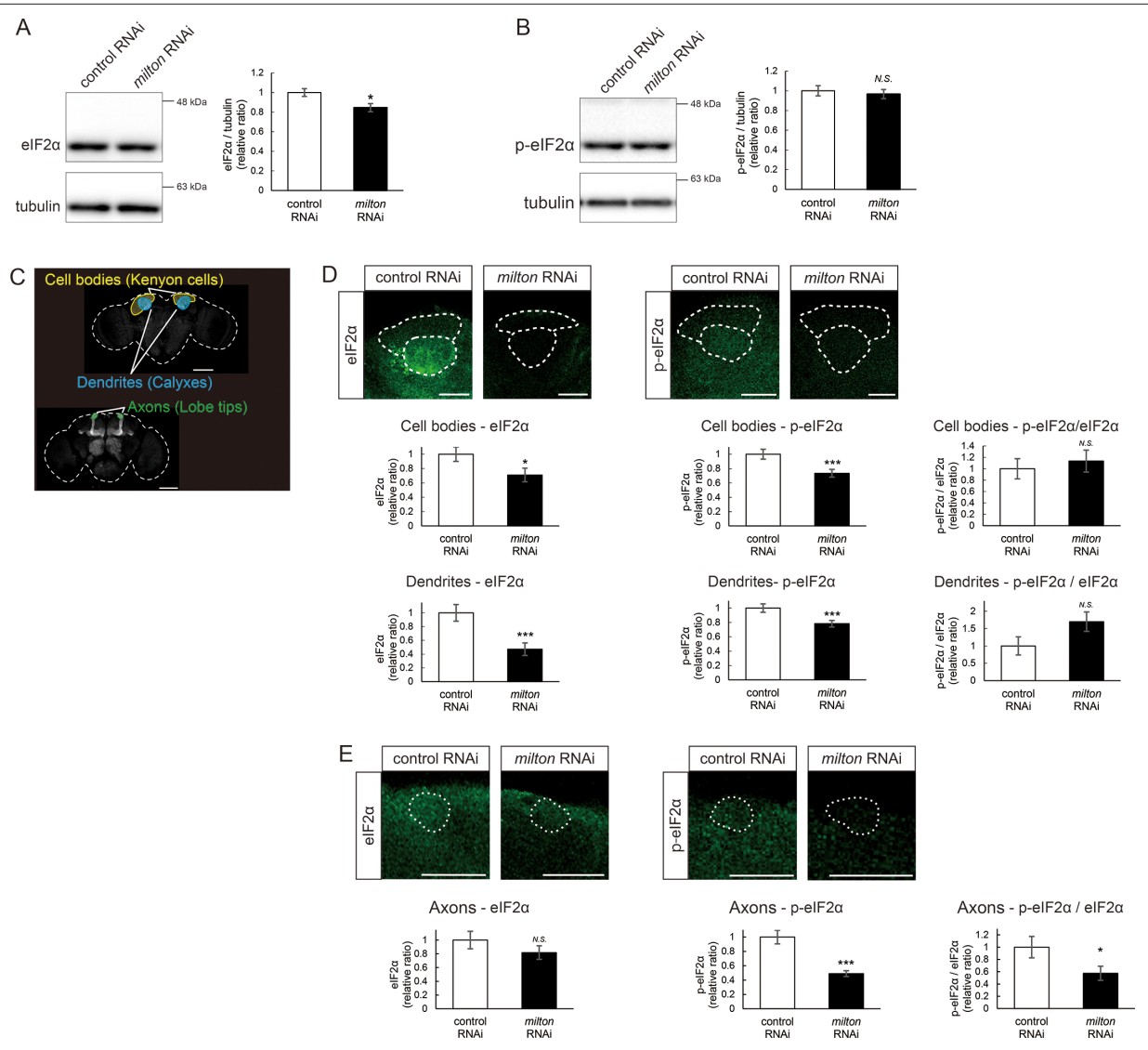

**Figure 5.** *milton* knockdown decreases phosphorylation of eIF2α. (**A, B**) Western blotting of head extracts with anti-eIF2α (**A**) and anti-p-eIF2α (**B**) antibodies. Flies were 14-day-old. Representative blots (left) and quantitation (right) are shown. Tubulin was used as a loading control. Means ± SE, n=6. (**C**) A schematic representation of the axon (Lobe tips), the cell body region (Kenyon cells), and dendritic region (Calyxes) in the fly brain. Scale bars, 100 μm. (**D, E**) Immunostaining with anti-eIF2α and anti-p-eIF2α antibodies. The mushroom body was identified by expression of mito-GFP. Scale bars, 20 μm. The signal intensities of eIF2α and p-eIF2α in axons, dendrites, and cell bodies were quantified and are shown as ratios relative to the control. Means ± SE, n=12. *N.S.*, p>0.05; *p<0.05; **p<0.01; ***p<0.005 (Student's *t*-test).

The online version of this article includes the following source data for figure 5:

**Source data 1.** PDF file containing original western blots for *Figure 5* indicating the relevant bands.

**Source data 2.** Original files for western blot analysis displayed in *Figure 5*.

## Lowering *eIF2β* rescues autophagic impairment and locomotor dysfunction induced by *milton* knockdown

Finally, we investigated whether suppression of eIF2β rescues autophagy impairment and locomotor dysfunction caused by neuronal knockdown of *milton*. Null mutants and flies with RNAi-mediated knockdown of *eIF2β* in neurons did not survive. Flies lacking one copy of the *eIF2β* gene survived without any gross abnormality, and the level of *eIF2β* mRNA in these flies was about 80% of that in control flies (*Figure 8A*). *eIF2β* heterozygosity did not affect the eIF2α and p-eIF2α levels (*Figure 8—figure supplement 1A and B*).

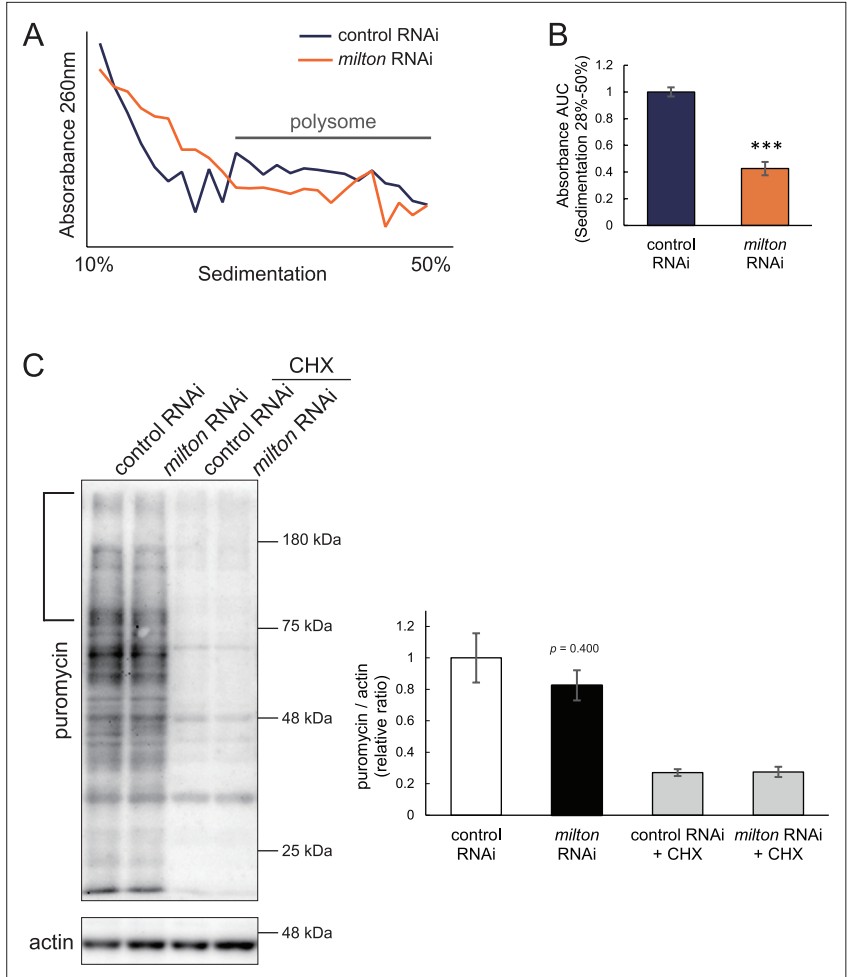

**Figure 6.** *milton* knockdown suppressed global translation. (**A**) Representative polysome traces of head lysates of control and *milton* knockdown flies. (**B**) Quantitation of polysome fraction. The relative ratio of area under the curve (AUC) of polysome fractions (sedimentation 28–50%). Means ± SE, n=3. ***p<0.005 (Student's *t*-test) (**C**) Western blotting of head lysates of control and *milton* knockdown flies fed puromycin alone or puromycin and cycloheximide (CHX) with an anti-puromycin antibody. Flies were 14-day-old. Actin was used as a loading control. Representative blots (left) and quantitation (right) are shown. Means ± SE, n=3. Student's *t*-test.

The online version of this article includes the following source data for figure 6:

**Source data 1.** PDF file containing original western blots for *Figure 6*, indicating the relevant bands.

**Source data 2.** Original files for western blot analysis displayed in *Figure 6*.

Neuronal knockdown of *milton* causes accumulation of autophagic substrate p62 in the Triton X-100-soluble fraction (*Figure 2B*), and we tested if lowering eIF2β ameliorates it. We found that *eIF2β* heterozygosity caused a mild increase in LC3-I levels and decreases in LC3-II levels, resulting in a significantly lower LC3-II/LC3-I ratio in *milton* knockdown flies (*Figure 8B*). *eIF2β* heterozygosity decreased the p62 level in the Triton X-100-soluble fraction in the brains of *milton* knockdown flies (*Figure 8C*). The p62 level in the SDS-soluble fraction, which is not sensitive to *milton* knockdown (*Figure 2B*), was not affected (*Figure 8C*). These results suggest that suppression of *eIF2β* ameliorates the impairment of autophagy caused by *milton* knockdown.

*eIF2β* heterozygosity also rescued locomotor dysfunction induced by *milton* knockdown. *milton* knockdown flies with *eIF2β* heterozygosity exhibited better locomotor function than *milton* knockdown alone (*Figure 8D*). The *milton* mRNA level was not increased in these flies, indicating that the rescue effect in the *eIF2β* heterozygous background was not mediated by an increase in the *milton* mRNA level (*Figure 8—figure supplement 1*). These data suggest that eIF2β upregulation mediates autophagy impairment and locomotor dysfunction caused by the depletion of axonal mitochondria.

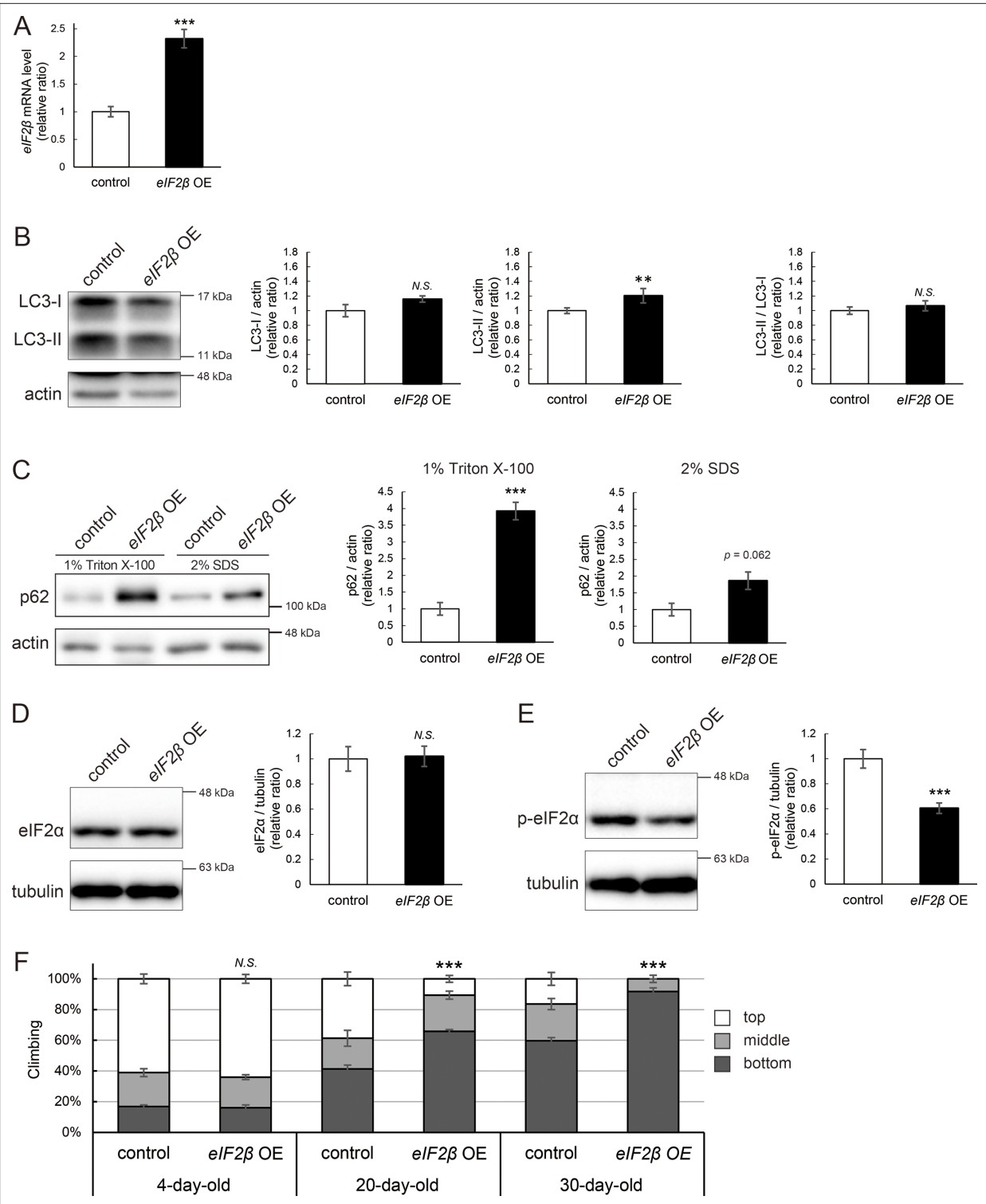

**Figure 7.** eIF2β upregulation impairs autophagy and decreases locomotor function. (**A**) *eIF2β* mRNA levels in head extracts of flies with UAS-*eIF2β* driven by elav-Gal4 (*eIF2β* OE) or UAS-GFP driven by elav-Gal4 (control) were quantified by qRT-PCR. Flies were 2-day-old. Means ± SE, n=4. (**B, C**) Western blotting of head extracts with anti-LC3 (**B**) and anti-p62 (**C**) antibodies. Flies were 14-day-old. Representative blots (left) and quantitation (right) are shown. Tubulin and actin were used as loading controls. Means ± SE, n=3 (p62), n=5 (LC3). (**D, E**) Western blotting of head extracts with anti-eIF2α (**D**) and anti-p-eIF2α (**E**) antibodies. Flies were 14-day-old. Representative blots (left) and quantitation (right) are shown. Tubulin was used as a loading control. Means ± SE, n=6. (**F**) Climbing assay revealed early-onset of age-dependent locomotor defects in *eIF2β*-overexpressing flies. Means ± SE, n=5. *N.S.*, p>0.05; ***p<0.005 (Student's *t*-test).

*Figure 7 continued on next page*

*Figure 7 continued*

The online version of this article includes the following source data and figure supplement(s) for figure 7:

**Source data 1.** PDF file containing original western blots for *Figure 7* indicating the relevant bands.

**Source data 2.** Original files for western blot analysis displayed in *Figure 7*.

**Figure supplement 1.** Histology analysis of fly heads with *eIF2β* overexpression.

## Discussion

The depletion of axonal mitochondria and accumulation of abnormal proteins are both characteristics of aged brains (*Currais et al., 2017*; *Grimm and Eckert, 2017*). Proteostasis perturbations trigger the formation of pathological aggregates and increase the risks of neurodegenerative diseases during aging. By using neuronal *milton* knockdown to deplete mitochondria from the axon, we provide evidence that loss of axonal mitochondria drives age-related proteostasis collapse via eIF2β (*Figure 9*). We observed declines in autophagy-mediated degradation of less-aggregated proteins and proteasome activity in *milton* knockdown flies (*Figure 2*). Accumulation of ubiquitinated proteins and changes in age-related pathways started prematurely in *milton* knockdown flies (*Figure 1* and *Table 2*). *milton* knockdown increased eIF2β and lowered eIF2α phosphorylation in young fly brain (*Figures 4 and 5*). Overexpression of *eIF2β* phenocopied the effects of *milton* knockdown, including reduced autophagy and accelerated age-related locomotor defects (*Figure 7*). Furthermore, lowering

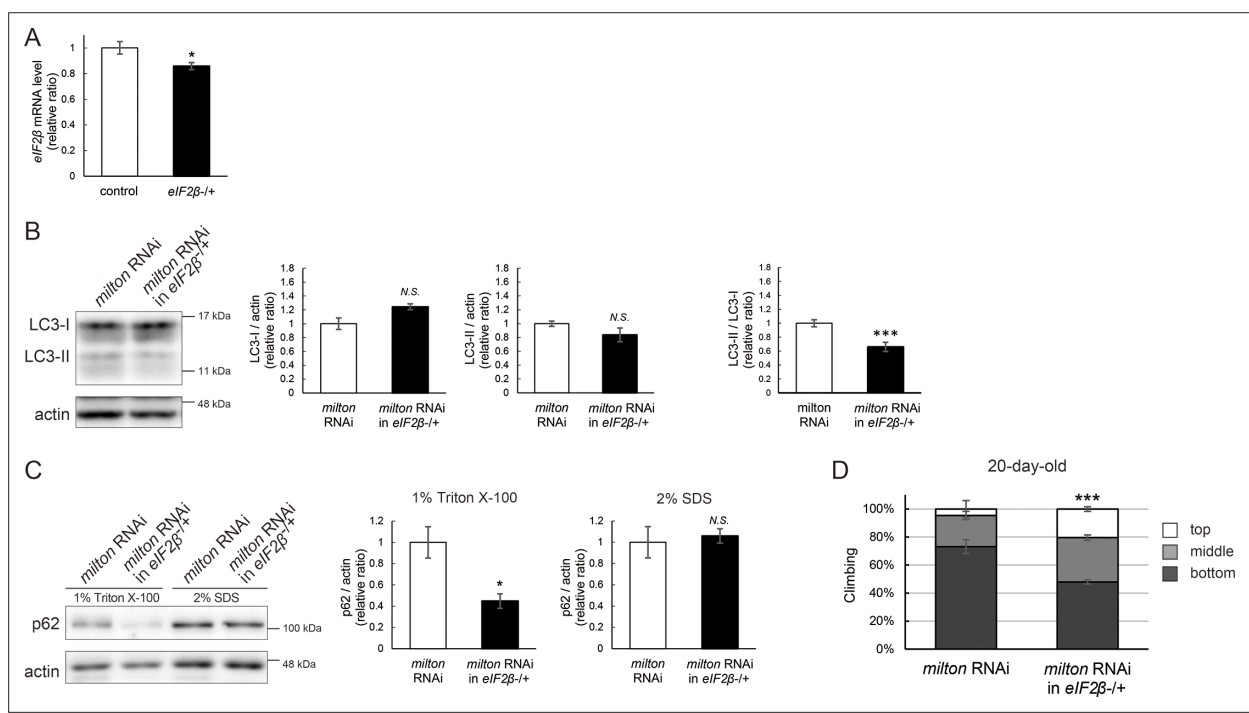

**Figure 8.** Lowering *eIF2β* rescues autophagic impairment and locomotor dysfunction induced by *milton* knockdown. (**A**) *eIF2β* mRNA levels with one disrupted copy of the *eIF2β* gene (*eIF2β*SAstopDsRed/+ [*eIF2β* -/+]). Head extracts of flies 2–3 day-old were analyzed by qRT-PCR. Means ± SE, n=3. (**B, C**) Western blotting of head extracts of flies with neuronal expression of *milton* RNAi with or without *eIF2β* heterozygosity with anti-LC3 (**B**) and anti-p62 (**C**) antibodies. Flies were 14-day-old. Representative blots (left) and quantitation (right) are shown. Actin was used as a loading control. Means ± SE, n=5 (LC3), n=3 (p62). (**D**) The climbing ability of 20-day-old flies expressing *milton* RNAi with or without *eIF2β* heterozygosity. Means ± SE, n=15. *N.S.*, p>0.05; *p<0.05; ***p<0.005 (Student's *t*-test).

The online version of this article includes the following source data and figure supplement(s) for figure 8:

**Source data 1.** PDF file containing original western blots for *Figure 8*, indicating the relevant bands.

**Source data 2.** Original files for western blot analysis displayed in *Figure 8*.

**Figure supplement 1.** Lowering the eIF2β level does not affect the levels of eIF2α and p-eIF2α.

**Figure supplement 1—source data 1.** PDF file containing original western blots for *Figure 8—figure supplement 1* indicating the relevant bands.

**Figure supplement 1—source data 2.** Original files for western blot analysis displayed in *Figure 8—figure supplement 1*.

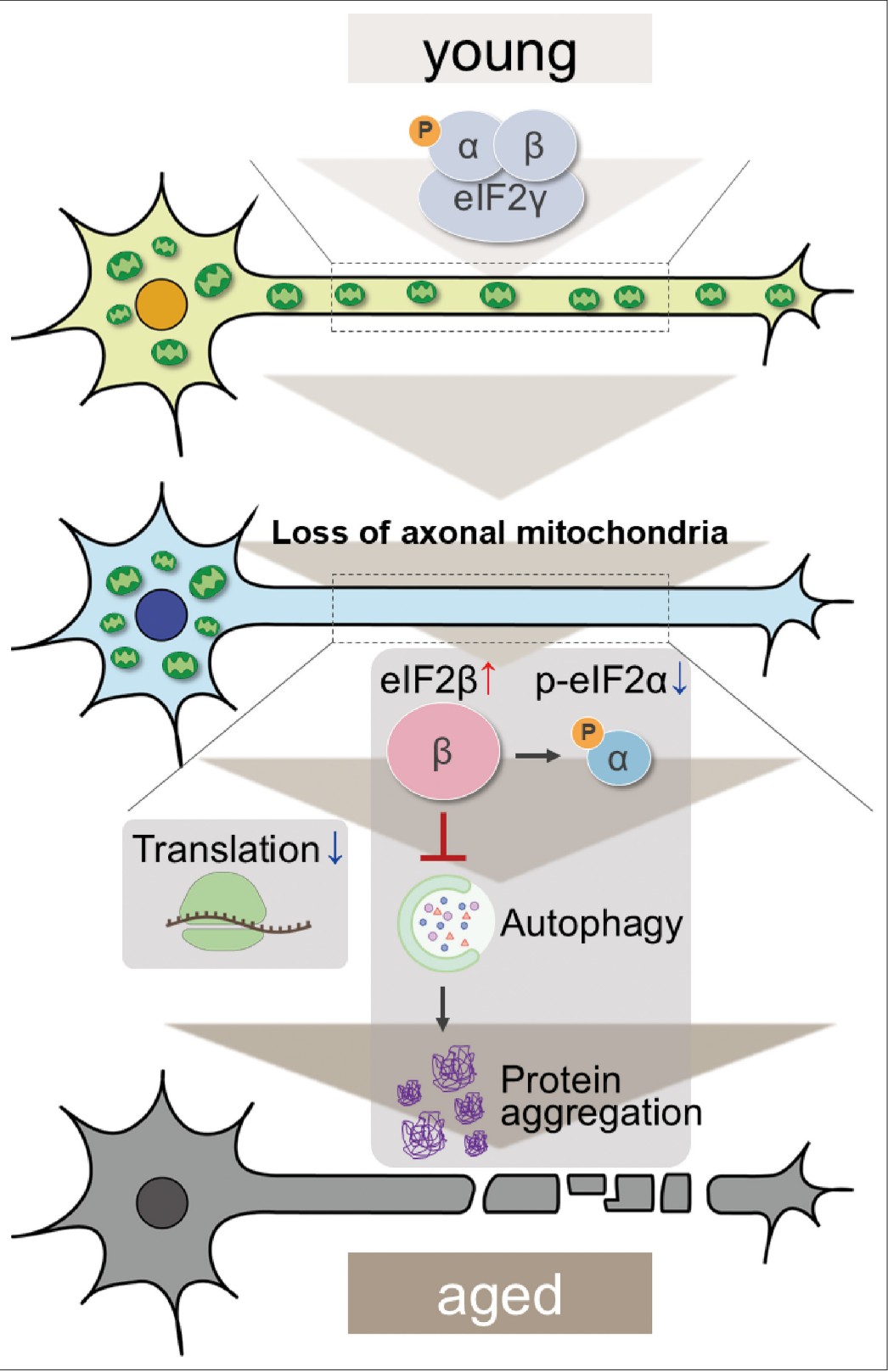

**Figure 9.** The mitochondria-eIF2β axis in the axon maintains neuronal proteostasis during aging. Aging is associated with a reduction in axonal transport of mitochondria. Our results suggest that the loss of axonal mitochondria leads to an increase in eIF2β, while the upregulation of eIF2β decreases autophagy-mediated protein degradation and promotes aging.

*eIF2β* levels suppressed the impairment of autophagy and locomotor dysfunction induced by *milton* knockdown (*Figure 8*). From these results, we propose that upregulation of eIF2β downstream of depletion of axonal mitochondria drives age-dependent collapse of proteostasis (*Figure 9*). Our results suggest that mitochondrial distribution and eIF2β are part of the mechanisms constituting proteostasis.

*milton* knockdown causes loss of mitochondria in the axon and accumulation of mitochondria in the soma. Thus, the detrimental effects may be mediated by the accumulation of mitochondria. However, degeneration induced by *milton* knockdown is prominent in the axon and not detected in the cell body (*Iijima-Ando et al., 2012*). Furthermore, abnormal protein accumulation was observed in the axon (*Figure 1*), and p-eIF2α/eIF2α was decreased in the neurites but not in the soma (*Figure 5*), suggesting that proteostasis defects studied in this work are caused by depletion of mitochondria rather than accumulation of mitochondria. Further analyses to dissect the effects of *milton* knockdown on proteostasis and translation in the cell body and axon by experiments with spatial resolution would be needed.

Our results suggest that the loss of axonal mitochondria is an event upstream of proteostasis collapse during aging. The number of puncta of ubiquitinated proteins was higher in milton knockdown at 14-day-old, but there was no significant difference at 30-day-old (*Figure 1*). Proteome analyses also showed that age-related pathways, such as immune responses, are enhanced in young flies with *milton* knockdown (*Table 2*). We also found that eIF2β protein levels increase in an age-dependent manner until 49-day-old and reduce after that (*Figure 4G*). In the brains with neuronal knockdown of *milton*, eIF2β levels were higher at 7 days old than those in control and lower at the 21 days old (*Figure 4D* and *Supplementary file 1*). These results suggest that *milton* knockdown is likely accelerating age-dependent changes rather than increasing their magnitude. Disruption of proteostasis is expected to contribute to neurodegeneration (*Grimm and Eckert, 2017*), and it would be interesting to analyze the sequence of protein accumulation and axonal degeneration in *milton* knockdown (*Iijima-Ando et al., 2012*; *Iijima-Ando et al., 2009* and *Figure 1*) in detail with higher time resolution.

Our results revealed that eIF2β regulates autophagy and maintains proteostasis during aging. eIF2β is a component of eIF2, which mediates translational regulation and ISR initiation. When ISR is activated, phosphorylated eIF2α suppresses global translation and induces translation of ATF4, which mediates transcription of autophagy-related genes (*Bond et al., 2020*; *B'chir et al., 2013*). Since ISR can positively regulate autophagy, we suspected that suppression of ISR underlies a reduction in autophagic protein degradation. We found neuronal knockdown of *milton* reduced phosphorylated eIF2α, suggesting that ISR is reduced (*Figure 5*). However, we also found that global translation was reduced (*Figure 6*). Increased levels of eIF2β might disrupt the eIF2 complex or alter its functions. The stoichiometric mismatch caused by an imbalance of eIF2 components may inhibit ISR induction. Supporting this model, we found that eIF2β upregulation reduced the levels of p-eIF2α (*Figure 7*). It is also possible that eIF2β mediates autophagy defects via mechanisms independent of ISR since eIF2β has functions independent of eIF2 (*Salton et al., 2017*; *Lee et al., 2007*). For example, suppression of *eIF2β* has been reported to slow down cancer cell growth (*Salton et al., 2017*). In developing neurons, eIF2β can directly interact with the translational repressor Kra to regulate midline axon guidance (*Lee et al., 2007*). Our results also suggest that milton knockdown and overexpression of eIF2β affect autophagy via increased LC3-I abundance (*Figures 2 and 7*), suggesting an unconventional mechanism of autophagy suppression. To our knowledge, the roles of eIF2β in aging and autophagy independent of ISR have not been reported. Our results revealed a novel function of eIF2β to maintain proteostasis during aging, while further investigation is required to elucidate underlying mechanisms.

How depletion of axonal mitochondria upregulates eIF2β is currently under investigation. A major mitochondrial function is ATP production, and depletion of axonal mitochondria downregulates ATP in axons (*Oka et al., 2021*). However, we found that ATP deprivation did not always suppress autophagy (*Figure 3*), suggesting it is unlikely to be involved in the mechanisms that induce eIF2β upregulation. Mitochondria also serve as signaling hubs for translation and protein degradation. Mitochondrial proteins are regulated by co-translational protein quality control, and mitochondrial damage induces translational stalling of mitochondrial outer membrane-associated *complex-I 30 kD subunit* (*C-I30*) mRNA (*Wu et al., 2019*). Additionally, the mitochondrial outer membrane ubiquitin ligase MITOL (also known as MARCHF5) ubiquitinates and regulates not only mitochondrial proteins such as Mfn2

(*Sugiura et al., 2013*) but also microtubule-associated (*Yonashiro et al., 2012*) and endoplasmic reticulum (*Takeda et al., 2019*) proteins. These findings indicate that mitochondria serve as local signaling centers for proteostasis maintenance, and eIF2β levels may also be regulated by mechanisms related to mitochondria.

In conclusion, our results suggest that axonal mitochondria and eIF2β form an axis to maintain constitutive autophagy. Suppression of *eIF2β* rescued autophagic defects and neuronal dysfunction upon loss of axonal mitochondria. Since eIF2β is conserved across many species, including *Drosophila* and humans, our results suggest that eIF2β may be a possible therapeutic target for aging and diseases associated with mitochondrial mislocalization.

# Materials and methods

**Key resources table**

| Reagent type (species) or resource | Designation | Source or reference | Identifiers | Additional information |
|---|---|---|---|---|
| Strain, strain background (*Drosophila*) | UAS-milton RNAi | Vienna *Drosophila* Resource Center (VDRC) | VDRC:v41508, FLYB:FBst0464139 | |
| Strain, strain background (*Drosophila*) | UAS-Miro RNAi | *Iijima-Ando et al., 2012* | | |
| Strain, strain background (*Drosophila*) | UAS-luciferase RNAi | *Iijima-Ando et al., 2012* | | |
| Strain, strain background (*Drosophila*) | UAS-Pfk RNAi | Bloomington *Drosophila* Stock Center | BDSC:36782, FLYB:FBti0146432 | |
| Strain, strain background (*Drosophila*) | UAS-luciferase RNAi | Bloomington *Drosophila* Stock Center | BDSC:31603, FLYB:FBti0130444 | |
| strain, strain background (*Drosophila*) | UAS-eIF2β | Bloomington *Drosophila* Stock Center | BDSC:17425, FLYB:FBti0038792 | |
| Strain, strain background (*Drosophila*) | UAS-GFP | Bloomington *Drosophila* Stock Center | BDSC:1521, FLYB:FBti0003040 | |
| Strain, strain background (*Drosophila*) | eIF2β[PBac{SAstopDsRed} LL07719] | KYOTO *Drosophila* Stock Center (DGRC) | DGRC:142114, FLYB:FBgn0004926 | |
| Strain, strain background (*Drosophila*) | w[1118] | Vienna *Drosophila* Resource Center (VDRC) | VDRC:60000 | |
| Strain, strain background (*Drosophila*) | UAS-mitoGFP | M. Saxton, University of California, Santa Cruz | | |
| Strain, strain background (*Drosophila*) | elav-GAL4 | Bloomington *Drosophila* Stock Center | BDSC:458, FLYB:FBti0002575 | |
| Strain, strain background (*Drosophila*) | GMR-gal4 | Bloomington *Drosophila* Stock Center | BDSC:1104, FLYB:FBti0002994 | |
| Antibody | anti-ubiquitin antibody Ubi-1 | Thermo Fisher | Cat#:13–1600, RRID:AB_2533002 | IHC:1:50 |
| Antibody | anti-LC3 antibody Atg8 | Merck Millipore | Cat#:ABC974, RRID:AB_2939040 | WB:1:1000 |
| Antibody | anti-p62 antibody Ref2P | Abcam | Cat#:ab178440, RRID:AB_2938801 | WB:1:750 |
| Antibody | anti-eIF2α | Abcam | Cat#:ab26197, RRID:AB_2096478 | IHC:1:50 |
| Antibody | anti-p-eIF2α | Cell signaling | Cat#:3398 S, RRID:AB_2096481 | IHC:1:50 |
| Antibody | anti-Drosophila eIF2β | This paper | | WB:1:1500 |
| Antibody | anti-puromycin | Enzo | Cat#:CAC-CAC-PEN-MA001, RRID:AB_2620162 | WB:1:1000 |

*Continued on next page*

*Continued*

| Reagent type (species) or resource | Designation | Source or reference | Identifiers | Additional information |
|---|---|---|---|---|
| Antibody | anti-actin | Sigma | Cat#:A2066, RRID:AB_476693 | WB:1:3000 |
| Antibody | anti-β tubulin | Sigma | Cat#:T9026, RRID:AB_477593 | WB:1:10000 |
| Antibody | peroxidase-conjugated goat anti-mouse IgG antibody | Dako | Cat#:P0447, RRID:AB_2617137 | WB:1:2000 |
| Antibody | peroxidase-conjugated pig anti-rabbit IgG antibody | Dako | Cat#:P0399, RRID:AB_2617141 | WB:1:2000 |
| Commercial Assay or Kit | 20 S Proteasome Substrate (SUC-LLVY-AMC) | Cayman | Cat#:10011095 | |
| Commercial Assay or Kit | ATP Determination Kit | Invitrogen | Cat#:A22066 | |

## Fly stocks and husbandry

Flies were maintained in standard cornmeal medium (10% glucose, 0.7% agar, 9% cornmeal, 4% yeast extract, 0.3% propionic acid, and 0.1% n-butyl p-hydroxybenzoate) at 25 °C under light–dark cycles of 12:12 hr. The flies were transferred to fresh food vials for every 2–3 days. UAS-*milton* RNAi (v41508) was from VDRC and outcrossed to [w1118] for five generations in our laboratory. Transgenic fly lines carrying UAS-*Miro* RNAi and UAS-*luciferase* RNAi (control for *milton* RNAi) were reported previously (*Iijima-Ando et al., 2012*). GMR-gal4, Elav-gal4, UAS-*Pfk* RNAi (Bloomington stock center #36782), UAS-*luciferase* RNAi (Bloomington stock center #31603) (control for *Pfk* RNAi), UAS-*GFP* (used for control for UAS-*eIF2β*), and UAS-*eIF2β* (eIF2β$^{EY08063}$, Bloomington stock center #17425) were from the Bloomington stock center. *eIF2β* loss-of-function strain (PBac{SAstopDsRed} LL07719, DGRC#142114) was from KYOTO *Drosophila* Stock Center. UAS-*mitoGFP* was a kind gift from Dr. W. M. Saxton (University of California, Santa Cruz). Fly genotypes used in this study are listed in *Supplementary file 2*.

## Immunohistochemistry and image acquisition

Fly brains were dissected in PBS and fixed for 45 min in formaldehyde (4% v/v in PBS) at room temperature. After incubation in PBST containing 0.1% Triton X-100 for 10 min three times, samples were incubated for 1 hr at room temperature in PBST containing 1% normal goat serum (Wako, #143–06561) and then incubated overnight with the primary antibody (anti-ubiquitin antibody Ubi-1 (Thermo Fisher #13–1600) (1:50), anti-eIF2α (abcam #ab26197) (1:50) and anti-p-eIF2α (Cell signaling #3398 S) (1:50)) diluted in 1% NGS/PBST at 4 °C. Samples were then washed for 10 min with PBST including 0.1% Triton X-100 three times and incubated with the secondary antibody overnight at 4 °C. Brains were mounted in Vectashield (Vectorlab Cat#H-1100) and analyzed under a confocal microscope (Nikon). Quantitative analysis was performed using ImageJ (National Institutes of Health) with maximum projection images derived from Z-stack images acquired with same settings. Puncta were identified with mean intensity and area using ImageJ. For eIF2α and p-eIF2α immunostaining, the mushroom body was detected by mitoGFP expression.

## Electron microscopy

Proboscis was removed from decapitated heads, which were then incubated in primary fixative solution (2.5% glutaraldehyde and 2% paraformaldehyde in 0.1 M sodium cacodylate buffer) at R.T. for 2 hr. After washing heads with 3% sucrose in 0.1 M sodium cacodylate buffer, fly heads were post-fixed for 1 hr in secondary fixation (1% osmium tetroxide in 0.1 M sodium cacodylate buffer) on ice. After washing with H$_2$O, heads were dehydrated with ethanol and infiltrated with propylene oxide and Epon mixture (TAAB and Nissin EM) for 3 hr. After infiltration, specimens were embedded with an Epon mixture at 70 °C for 2–3 days. Thin sections (70 nm) of laminas were collected on copper grids. The sections were stained with 5% uranyl acetate in 50% ethanol and Reynolds' lead citrate

solution. Electron micrographs were obtained with a CCD camera mounted on a JEM-1400 plus electron microscope (Jeol Ltd.). Quantitation was performed using ImageJ (National Institutes of Health).

## SDS–PAGE and immunoblotting

Western blotting was performed as reported previously (*Iijima-Ando et al., 2012*). Briefly, heads of 10–20 *Drosophila* were homogenized with SDS-Tris-Glycine sample buffer (0.312 M Tris, 5% SDS, 8% glycerol, 0.0625% BPB, 10% β-mercaptoethanol, 10 µg/mL leupeptin, 0.4 µM Pefabloc, 10 mM β-glycerophosphate, 10 mM NaF) and after boiling at 95 °C for 2 min, it was centrifuged at 13,200 rpm, and the supernatant was used as a sample. For p62 western blot, fly heads were homogenized with 1% PBST and after centrifugation at 13,200 rpm, the supernatant was mixed 1:1 SDS-Tris-Glycine sample buffer, and boiled at 95 °C for 2 min. The pellet was dissolved with 2% SDS in PBS, then centrifuged again at 13,200 rpm. The supernatant was mixed 1:1 SDS-Tris-Glycine sample buffer and then boiled at 95 °C for 2 min. SDS–PAGE for western blotting was performed using 15%(w/v) (LC3), 7.5%(w/v) (p62), 10% (w/v) (eIF2α, β, and p-eIF2α) polyacrylamide gels. After electrophoresis, they were transferred to PVDF membrane (Merck Millipore) using a transfer device (BIO-RAD). After transfer, the membrane was blocked with 5% skim milk/TBST (50 mM Tris (pH 7.5), 0.15 M NaCl, 0.05% Tween20) for 1 hr and incubated with primary antibody listed below overnight at 4 °C. Membranes were rinsed twice with TBST containing 0.65 M NaCl and once with TBST containing 0.15 M NaCl. After incubation with the secondary antibody at room temperature for 1 hr, membranes were rinsed twice with TBST containing 0.65 M NaCl and once with TBST containing 0.15 M NaCl. After incubation with Immobilon Western Chemiluminescent HRP Substrate (Merck Millipore), chemiluminescent signals were detected with Fusion FX (Vilber). Experiments were repeated at least three times with independent cohorts of flies.

### Primary antibodies

anti-LC3 antibody Atg8 (Merck Millipore #ABC974) (1:1000), anti-p62 antibody Ref2P (Abcam #ab178440) (1:750), anti-eIF2β antibody (1:1500), anti-eIF2α antibody (Abcam #ab26197) (1:1000), anti-p-eIF2α antibody (Cell signaling #3398 S) (1:2000), anti-actin antibody (Sigma #A2066) (1:3000), and anti-β tubulin antibody (Sigma #T9026) (1:100,000). Polyclonal anti-eIF2β antibody was raised against a synthetic peptide (CGLEDDTKKEDPQDEA) corresponding to the C-terminal residues 29–43 of *Drosophila* eIF2β (1:1500).

### Secondary antibodies

Peroxidase-conjugated goat anti-mouse IgG antibody (Dako #P0447) (1:2000), peroxidase-conjugated pig anti-rabbit IgG antibody (Dako #P0399) (1:2000).

## Proteasome assay

Heads from ten flies were homogenized in 150 µl of buffer B (25 mM Tris-HCl [pH 7.5], 2 mM ATP, 5 mM MgCl2, and 1 mM dithiothreitol). Proteasome peptidase activity in the lysates was measured with a synthetic peptide substrate, succinyl-Leu-Leu-Val-Tyr-7-amino-4-methyl-coumarin (Suc-LLVY-AMC; Cayman). Luminescence was measured on a multimode plate reader 2300 Enspire (PerkinElmer). Experiments were repeated at least three times with independent cohorts of flies.

## ATP assay

Heads from the 10 flies were homogenized in 50 µl of 6 M guanidine-HCl in extraction buffer (100 mM Tris and 4 mM EDTA, pH 7.8) to inhibit ATPases. Samples were boiled for 5 min and centrifuged. The supernatant was diluted 4% with extraction buffer and mixed with a reaction solution (ATP Determination kit, Invitrogen). Luminescence was measured on a multimode plate reader 2300 Enspire (PerkinElmer). The relative ATP levels were calculated by dividing the luminescence by the total protein concentration, which was determined by the Bradford method. Experiments were repeated at least three times with independent cohorts of flies.

## Proteomic assay and pathway analysis

### Sample preparation

Heads from the 35 flies were homogenized in 110 µl of extraction buffer (0.25% RapiGest SF, 50 mM ammonium bicarbonate, 10 mM dithiothreitol, 10 µg/mL leupeptin, 0.4 µM Pefabloc, 10 mM

β-glycerophosphate, 10 mM NaF). Homogenized samples were centrifuged and boiled for 5 min. After quantification of the protein concentration using a Pierce 660 nm Protein Assay (Thermo Fisher Scientific), 10 μg proteins from each sample were reduced using 5 mM tris (2-carboxyethyl) phosphine hydrochloride (TCEP-HCl; Thermo Fisher Scientific) at 60 °C for 1 hr, alkylated using 15 mM iodoacetamide (Fujifilm Wako Pure Chemical, Osaka, Japan) at room temperature for 30 min, and then digested using 1.5 μg Trypsin Gold (Mass Spectrometry Grade; Promega, Madison, WI, USA) at 37 °C for 17 hr. The digests were acidified by the addition of trifluoroacetic acid (TFA), incubated at 37 °C for 30 min, and then centrifuged at 17,000×$g$ for 10 min to remove the RapiGest SF. The supernatants were collected and desalted using MonoSpin C18 (GL Sciences, Tokyo, Japan). The resulting eluates were concentrated *in vacuo*, dissolved in 2% MeCN containing 0.1% formic acid (FA), and subjected to LC-MS/MS analysis.

## LC-MS/MS analysis and database search

LC-MS/MS analyses were performed on an Ultimate 3000 RSLCnano system (Thermo Fisher Scientific) coupled to a Q Exactive hybrid quadrupole-Orbitrap mass spectrometer (Thermo Fisher Scientific) equipped with a nano electron spray ionization (ESI) source. The LC system was equipped with a trap column (C18 PepMap 100, 0.3×5 mm, 5 μm, Thermo Fisher Scientific) and an analytical column (NTCC-360/75-3-125, Nikkyo Technos, Tokyo, Japan). Peptide separation was performed using a 90 min gradient of water/0.1% FA (mobile phase A) and MeCN/0.1% FA (mobile phase B) at a flow rate of 300 nL/min. Elution was performed as follows: 0–3 min, 2% B; 3–93 min, 2–40% B; 93–95 min, 40–95% B; 95–105 min, 95% B; 105–107 min, 95–2% B; and 107–120 min, 2% B. The mass spectrometer was operated in data-dependent acquisition mode. The MS parameters were as follows: spray voltage, 2.0 kV; capillary temperature, 275 °C; S-lens RF level, 50; scan type, full MS; scan range, $m/z$ 350–1500; resolution, 70,000; polarity, positive; automatic gain control target, 3×10$^6$; and maximum injection time, 100 ms. The MS/MS parameters were as follows: resolution, 17,500; automatic gain control target, 1×10$^5$; maximum injection time, 60 msec; normalized collision energy (NCE), 27; dynamic exclusion, 15 s; loop count, 10; isolation window, 1.6 $m/z$; charge exclusion: unassigned, 1 and ≥8; and injection volume, 1 μL (containing 0.5 μg protein). Measurements were made in duplicate for each sample.

The identification of proteins and label-free quantification (LFQ) of the detected peptides was performed using Proteome Discoverer software ver. 2.4 (Thermo Fisher Scientific). The analytical parameters used for the database search were as follows: parent mass error tolerance, 10.0 ppm; fragment mass error tolerance, 0.02 Da; search engine, sequest HT; protein database, *Drosophila melanogaster* (Fruit fly: SwissProt Tax ID = 7227); enzyme name, trypsin (full); maximum number of missed cleavages, 2; dynamic modification, oxidation (methionine), phosphorylation (serine, threonine, tyrosine), acetyl (lysine), GG (lysine); N-terminal modification, Met-loss (methionine), and Met-loss+acetyl (methionine); static modification, carbamidomethylation (cysteine) and FDR confidence, High <0.01, 0.01 ≤ Medium < 0.05, 0.05 ≤ Low. The parameters for LFQ were as follows: precursor abundance, based on area; and normalization mode, total peptide amount.

The abundance ratio of *milton* RNAi to control RNAi at 7- or 21-day-old was calculated. We considered proteins with an abundance ratio of ≥2.0 or≤0.5 and an ANOVA p-value of <0.05 based on volcano plots to be differentially expressed of *milton* RNAi. To extract molecular networks biologically relevant to the proteins that are differentially expressed in *milton* RNAi, pathway analysis was performed using KeyMolnet (KM Data Inc, Tokyo, Japan).

## RNA extraction and quantitative real-time PCR analysis

Heads from more than 25 flies were mechanically isolated, and total RNA was extracted using ISOGEN (NipponGene) followed by reverse-transcription using PrimeScript RT reagent kit (Takara). The resulting cDNA was used as a template for PCR with THUNDERBIRD SYBR qPCR mix (TOYOBO) on a Thermal Cycler Dice real-time system TP800 (Takara). Expression of genes of interest was standardized relative to rp49. Relative expression values were determined by the ΔΔCT method. Experiments were repeated three times, and a representative result was shown.

Primers were designed using DRSC FlyPrimerBank (Harvard Medical School). Primer sequences are shown below:

*eIF2β* for 5′-GGACGACGACAAGAGCGAAG-3′

*eIF2β* rev 5′-CGGTCGCATCACGAACTTTG-3′
*milton* for 5′-GGCTTCAGGGCCAGGTATCT-3′
*milton* rev 5′-GCCGAACTTGGCTGACTTTG-3′
*Actin* for 5′-TGCACCGCAAGTGCTTCTAA-3′
*Actin* rev 5′-TGCTGCACTCCAAACTTCCA-3′
*rp49* for 5′-GCTAAGCTGTCGCACAAATG-3′
*rp49* rev 5′- GTTCGATCCGTAACCGATGT-3′

## Polysome gradient centrifugation

30 heads were homogenized in 150 µl of lysis buffer (25 mM Tris pH 7.5, 50 mM MgCl2, 250 mM NaCl, 1 mM DTT, 0.5 mg/ml cycloheximide, 0.1 mg/ml heparin). The lysates were centrifuged at 13,200 rpm at 4 °C for 5 min, and the supernatant was collected. The samples containing 38 µg of RNA were layered gently on top of a 10–50% w/w sucrose gradient (50 mM Tris pH 7.5, 50 mM MgCl2, 250 mM NaCl, 0.1 mg/ml heparin, 0.5 mg/ml cycloheximide in 5 ml polyallomer tube) and centrifuged at 37,000 rpm at 4 °C for 150 min in a himac CP-NX ultracentrifuge using a P50AT rotor. Samples were fractionated from top to bottom, and absorbance at OD260 nm was analyzed by a Plate reader (EnSpire). Experiments were repeated at least three times with independent cohorts of flies.

## Puromycin analysis

13-day-old flies were starved for 6 hr and fed 600 µM puromycin (Sigma) or 600 µM puromycin/35 mM cycloheximide (Sigma) in 5% sucrose solution for 20 hr. Incorporated puromycin was quantified by western blot with anti-puromycin antibody (Enzo # CAC-CAC-PEN-MA001) and normalized with actin. Experiments were repeated at least three times with independent cohorts of flies.

## Histological analysis

Fly heads were fixed in Bouin's fixative solution for 48 hr at room temperature, incubated for 24 hr in 50 mM Tris/150 mM NaCl, and embedded in paraffin. Serial sections (7 µm thickness) through the entire heads were stained with hematoxylin and eosin and examined by bright-field microscopy. Images of the sections that include the lamina were captured with Keyence microscope BZ-X700 (Keyence), and the vacuole area was measured using ImageJ (National Institutes of Health).

## Climbing assay

The climbing assay was performed as previously described (*Iijima-Ando et al., 2012*). Flies were placed in an empty plastic vial (2.5 cm in diameter ×10 cm in length). The vial was gently tapped to knock the flies to the bottom, and the number of flies that reached the top, middle, and bottom areas of the vials in 10 s was counted. Experiments were repeated 10 times, and the mean percentage of flies in each area and standard deviations were calculated. Experiments were repeated with independent cohorts more than three times, and a representative result was shown.

## Statistics

The number of replicates, what n represents, precision measurements, and the meaning of error bars are indicated in Figure Legends. Data are shown as means ± SEM. For pairwise comparisons, Student's t-test was performed with Microsoft Excel (Microsoft). For multiple comparisons, data were analyzed using one-way ANOVA with Tukey's HSD multiple-comparisons test in the GraphPad Prism 6.0 software (GraphPad Software, Inc, La Jolla, CA). Results with a p-value of less than 0.05 were considered to be statistically significant.

## Acknowledgements

The authors thank the Bloomington stock center; TRiP at Harvard Medical School (NIH/NIGMS R01-GM084947); the Kyoto Drosophila Stock Center and the Vienna Drosophila RNAi Center for fly stocks. The authors thank Dr. Masayuki Miura from Department of Pharmaceutical Science, University of Tokyo, for proteasome activity assay protocol; Dr. Shin-ichi Hisanaga from the Department of Biological Sciences, Tokyo Metropolitan University, for critical comments; Dr. Taro Saito, Dr. Akiko Asada from the Department of Biological Sciences, Tokyo Metropolitan University, Dr. Michiko Sekiya from

Department of Alzheimer's Disease Research, National Center for Geriatrics and Gerontology, and Dr. Seiji Watanabe, Dr. Koji Yamanaka from Department of Neuroscience and Pathobiology, Research Institute of Environmental Medicine, Nagoya University for technical supports. This work was supported by the Sasakawa Scientific Research Grant (2021-4087) (to KS), the Takeda Science Foundation (to KA), Hoansha foundation grant (to KA), a research award from the Japan Foundation for Aging and Health (to KA), the Novartis Foundation (Japan) for the promotion of Science (to KA), JSPS KAKENHI Grant-in-Aid for Scientific Research on Challenging Research (Exploratory) JP19K21593 (to KA), JSPS KAKENHI Grant-in-Aid for Scientific Research(B) JP24K02860 (to KA), NIG-JOINT (National Institute of Genetics, 71A2018, 25A2019) (to KA) and TMU strategic research fund for social engagement (to KA).

## Additional information

### Funding

| Funder | Grant reference number | Author |
|---|---|---|
| Japan Science Society | Sasakawa Scientific Research Grant (2021-4087) | Kanako Shinno |
| Takeda Science Foundation | | Kanae Ando |
| Hoansha Foundation | | Kanae Ando |
| Japan Foundation for Aging and Health | | Kanae Ando |
| NOVARTIS Foundation | | Kanae Ando |
| Japan Society for the Promotion of Science | JP19K21593 | Kanae Ando |
| Japan Society for the Promotion of Science | JP24K02860 | Kanae Ando |
| National Institute of Genetics | NIG-JOINT 71A2018 | Kanae Ando |
| National Institute of Genetics | NIG-Joint 25A2019 | Kanae Ando |
| Tokyo Metropolitan University | TMU strategic research fund for social engagement | Kanae Ando |

The funders had no role in study design, data collection and interpretation, or the decision to submit the work for publication.

### Author contributions

Kanako Shinno, Conceptualization, Resources, Formal analysis, Funding acquisition, Investigation, Visualization, Writing – original draft, Writing – review and editing; Yuri Miura, Resources, Formal analysis, Supervision, Investigation, Visualization, Methodology, Writing – review and editing; Koichi M Iijima, Resources; Emiko Suzuki, Resources, Formal analysis, Supervision, Methodology, Writing – review and editing; Kanae Ando, Conceptualization, Resources, Formal analysis, Supervision, Funding acquisition, Investigation, Writing – original draft, Project administration, Writing – review and editing

### Author ORCIDs

Kanako Shinno ⓘ https://orcid.org/0009-0002-7114-7686
Yuri Miura ⓘ https://orcid.org/0000-0003-1239-3780
Koichi M Iijima ⓘ https://orcid.org/0000-0003-4794-1863
Emiko Suzuki ⓘ https://orcid.org/0000-0002-4005-0542
Kanae Ando ⓘ https://orcid.org/0000-0002-3956-276X

Reviewer #1 (Public review): https://doi.org/10.7554/eLife.95576.5.sa1
Reviewer #2 (Public review): https://doi.org/10.7554/eLife.95576.5.sa2
Author response https://doi.org/10.7554/eLife.95576.5.sa3

## Additional files

### Supplementary files

Supplementary file 1. Excel file containing a list of proteins detected by liquid chromatography-tandem mass spectrometry (LC-MS/MS).

Supplementary file 2. Excel file containing a list of fly genotypes used in this study.

MDAR checklist

### Data availability

The datasets used and/or analyzed in the current study are available in jPOST (https://rep-demo.jpostdb.org/) with jPOST ID: JPDM000120.

The following dataset was generated:

| Author(s) | Year | Dataset title | Dataset URL | Database and Identifier |
|---|---|---|---|---|
| Shinno K | 2025 | Axonal distribution of mitochondria maintains neuronal autophagy during aging | https://rep-demo.jpostdb.org/entry/JPDM000120 | JPOST, JPDM000120 |

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
