## [Editor Report · eLife Assessment]

In flies defective for axonal transport of mitochondria, the authors report the upregulation of one subunit, the beta subunit, of the heterotrimeric eIF2 complex via mass spectroscopy proteomics. Neuronal overexpression of eIF2β phenocopied aspects of neuronal dysfunction observed when axonal transport of mitochondria was compromised. Conversely, lowering eIF2β expression suppressed aspects of neuronal dysfunction. While these are intriguing and **useful** observations, technical weaknesses limit the interpretation. On balance, the evidence supporting the current claims is suggestive but **incomplete**, especially concerning the characterization of the eIF2 heterotrimer and the data regarding translational regulation.

---

## [Referee Report · Reviewer #1 (Public review)]

The study presents significant findings on the role of mitochondrial depletion in axons and its impact on neuronal proteostasis. It effectively demonstrates how the loss of axonal mitochondria and elevated levels of eIF2β contribute to autophagy collapse and neuronal dysfunction. The use of Drosophila as a model organism and comprehensive proteome analysis adds robustness to the findings.

In this revision, the authors have responded thoughtfully to previous concerns. In particular, they have addressed the need for a quantitative analysis of age-dependent changes in eIF2β and eIF2α. By adding western blot data from multiple time points (7 to 63 days), they show that eIF2β levels gradually increase until middle age, then decline. In milton knockdown flies, this pattern appears shifted, supporting the idea that mitochondrial defects may accelerate aging-related molecular changes. These additions clarify the temporal dynamics of eIF2β and improve the overall interpretation.

Other updates include appropriate corrections to figures and quantification methods. The authors have also revised some of their earlier mechanistic claims, presenting a more cautious interpretation of their findings.

Overall, this work provides new insights into how mitochondrial transport defects may influence aging-related proteostasis through eIF2β. The manuscript is now more convincing, and the revisions address the main points raised earlier. I find the updated version much improved.

---

## [Referee Report · Reviewer #2 (Public review)]

In the manuscript, the authors aimed to elucidate the molecular mechanism that explains neurodegeneration caused by the depletion of axonal mitochondria. In Drosophila, starting with siRNA depletion of milton and Miro, the authors attempted to demonstrate that the depletion of axonal mitochondria induces the defect in autophagy. From proteome analyses, the authors hypothesized that autophagy is impacted by the abundance of eIF2β and the phosphorylation of eIF2α. The authors followed up the proteome analyses by testing the effects of eIF2β overexpression and depletion on autophagy. With the results from those experiments, the authors proposed a novel role of eIF2β in proteostasis that underlies neurodegeneration derived from the depletion of axonal mitochondria, which they suggest accelerates age-dependent changes rather than increasing their magnitude.

Strong caution is necessary regarding the interpretation of translational regulation resulting from the milton KD. The effect of milton KD on translation appears subtle, if present at all, in the puromycin incorporation experiments in both the initial and revised versions. Additionally, the polysome profiling data in the revised manuscript lack the clear resolution for ribosomal subunits, monosomes, and polysomes that is typically expected in publications.

---

## [Author Response]

The following is the authors’ response to the previous reviews

**Reviewer #1 (Public review):**
The study presents significant findings on the role of mitochondrial depletion in axons and its impact on neuronal proteostasis. It effectively demonstrates how the loss of axonal mitochondria and elevated levels of eIF2β contribute to autophagy collapse and neuronal dysfunction. The use of Drosophila as a model organism and comprehensive proteome analysis adds robustness to the findings.In this revision, the authors have responded thoughtfully to previous concerns. In particular, they have addressed the need for a quantitative analysis of age-dependent changes in eIF2β and eIF2α. By adding western blot data from multiple time points (7 to 63 days), they show that eIF2β levels gradually increase until middle age, then decline. In milton knockdown flies, this pattern appears shifted, supporting the idea that mitochondrial defects may accelerate aging-related molecular changes. These additions clarify the temporal dynamics of eIF2β and improve the overall interpretation.Other updates include appropriate corrections to figures and quantification methods. The authors have also revised some of their earlier mechanistic claims, presenting a more cautious interpretation of their findings.Overall, this work provides new insights into how mitochondrial transport defects may influence aging-related proteostasis through eIF2β. The manuscript is now more convincing, and the revisions address the main points raised earlier. I find the updated version much improved.

Thank you so much for the review, insightful comments and encouragement. We appreciate it.

**Reviewer #2 (Public review):**
In the manuscript, the authors aimed to elucidate the molecular mechanism that explains neurodegeneration caused by the depletion of axonal mitochondria. In Drosophila, starting with siRNA depletion of milton and Miro, the authors attempted to demonstrate that the depletion of axonal mitochondria induces the defect in autophagy. From proteome analyses, the authors hypothesized that autophagy is impacted by the abundance of eIF2β and the phosphorylation of eIF2α. The authors followed up the proteome analyses by testing the effects of eIF2β overexpression and depletion on autophagy. With the results from those experiments, the authors proposed a novel role of eIF2β in proteostasis that underlies neurodegeneration derived from the depletion of axonal mitochondria, which they suggest accelerates age-dependent changes rather than increasing their magnitude.Strong caution is necessary regarding the interpretation of translational regulation resulting from the milton KD. The effect of milton KD on translation appears subtle, if present at all, in the puromycin incorporation experiments in both the initial and revised versions. Additionally, the polysome profiling data in the revised manuscript lack the clear resolution for ribosomal subunits, monosomes, and polysomes that is typically expected in publications.

Thank you so much for the review and insightful comments. We appreciate it.

**Reviewer #2 (Recommendations for the authors):**
The revised manuscript demonstrates many improvements. The authors have provided a more comprehensive data set and a more detailed description of their results. Furthermore, their explanation of the Integrated Stress Response (ISR) has been corrected, and this correction is reflected in the data interpretation.As in the public review, I maintained my emphasis on the weakness of the claim on suppressed global translation, since the data are the same in the initial and the revised versions.

Thank you for your review. We understand that further studies will be needed to elucidate the roles on mitochondrial distribution in global translation profile. We will keep working on it.

A few suggestions for minor corrections.(1) The order of figures in the revised version is disorganized.

Thank you for pointing it out. We corrected the order.

(2) In Figure 1A, mitochondria is bound by milton, and kinesin is bound by Miro. Their roles should be opposite.

Thank you for pointing it out, and we are sorry for the oversight. We corrected it.